# A Quantum-like Model of Interdependence for Embodied Human–Machine Teams: Reviewing the Path to Autonomy Facing Complexity and Uncertainty

**DOI:** 10.3390/e25091323

**Published:** 2023-09-11

**Authors:** William F. Lawless, Ira S. Moskowitz, Katarina Z. Doctor

**Affiliations:** 1Department of Mathematics and Psychology, Paine College, Augusta, GA 30901, USA; 2Naval Research Laboratory, Information Technology Division, Washington, DC 20375, USA; ira.moskowitz@nrl.navy.mil (I.S.M.); katarina.doctor@nrl.navy.mil (K.Z.D.)

**Keywords:** interdependence, human–machine teams, autonomy, complexity, problem tasks, entropy, embodied cognition, harmonic oscillators

## Abstract

In this review, our goal is to design and test quantum-like algorithms for Artificial Intelligence (AI) in open systems to structure a human–machine team to be able to reach its maximum performance. Unlike the laboratory, in open systems, teams face complexity, uncertainty and conflict. All task domains have complexity levels—some low, and others high. Complexity in this new domain is affected by the environment and the task, which are both affected by uncertainty and conflict. We contrast individual and interdependence approaches to teams. The traditional and individual approach focuses on building teams and systems by aggregating the best available information for individuals, their thoughts, behaviors and skills. Its concepts are characterized chiefly by one-to-one relations between mind and body, a summation of disembodied individual mental and physical attributes, and degrees of freedom corresponding to the number of members in a team; however, this approach is characterized by the many researchers who have invested in it for almost a century with few results that can be generalized to human–machine interactions; by the replication crisis of today (e.g., the invalid scale for self-esteem); and by its many disembodied concepts. In contrast, our approach is based on the quantum-like nature of interdependence. It allows us theorization about the bistability of mind and body, but it poses a measurement problem and a non-factorable nature. Bistability addresses team structure and performance; the measurement problem solves the replication crisis; and the non-factorable aspect of teams reduces the degrees of freedom and the information derivable from teammates to match findings by the National Academies of Science. We review the science of teams and human–machine team research in the laboratory versus in the open field; justifications for rejecting traditional social science while supporting our approach; a fuller understanding of the complexity of teams and tasks; the mathematics involved; a review of results from our quantum-like model in the open field (e.g., tradeoffs between team structure and performance); and the path forward to advance the science of interdependence and autonomy.

## 1. Introduction

In this review, our goal [1] is to enable autonomous human–machine teams (AHMTs) to be designed with a set of algorithms that can be deployed with Artificial Intelligence (AI) to structure a human–machine team able to reach its maximum performance in the open, specifically away from the laboratory where most team science is currently practiced. In the open, but not in the laboratory, teams face uncertainty and conflict where, as Mann [2] has concluded, rational beliefs fail. We begin with a historical review of interdependence from a computational perspective.

Then, in background sections, (1) we review and define interdependence; (2) we justify why social science has failed to produce a mathematics of coherence in teams (here, coherence describes a state of interdependence that has not been corrupted or interrupted: for example, see “redundancy” below); (3) we review the effect of task domain complexity to determine whether there is also an effect of interdependence in this domain besides the effect of domain complexity; and (4) we justify our quantum model of interdependence based on the interdependence that exists among teammates. In the body of our article, we review what we know of the mathematics of interdependence in teams, what we know so far of its generalizations (e.g., how to exploit them), and some of the barriers remaining to make autonomous human–machine teams a new discipline for the science of interdependence.

### 1.1. Background: Definitions of Interdependence

To build and operate systems of autonomous human–machine teams characterized by interdependence requires an operational definition of interdependence and its effects that both humans and machines can perceive, interpret and address in open contexts. Below, we introduce a definition of interdependence from a historical and computational perspective (Table 1). We end with our proposed working definition of interdependence. In the next three background subsections, we justify moving away from traditional social science, as in the molds of Spinoza and Hume; we consider the effects of domain complexity on interdependence; and we justify our quantum-like (QL) model of interdependence. Next, we review our mathematical approach to interdependence.

Interdependence drives the interaction between two agents [5]; it is the social field that holds teams together or breaks them apart [3] and it gives a team’s teammates power over what an equal number of individuals working independently of each other can accomplish [14]. However, without a formal, computational approach to the study of teams, autonomous human–machine teams are ad hoc, vulnerable to those with the intuition of what makes a team potentially and significantly more powerful than an equal collection of the same individuals in a team acting independently of each other. Uniformly across the social sciences, the traditional assumption has been based on Spinoza’s and Hume’s traditional, non-dual mind–body equivalence which prevents any affordance for the power derived from a group constituted as a team [14].

We contrast this traditional approach with our computational approach based on the dualism of the mind–body, the two equivalent, competing, and countering centers that the human brain draws upon. In our approach, the tension between the two countering views of reality, one by the mind and the other embodied in reality [15], a form of dualism, provide a means to sharply focus the information derived from a situation to computationally determine and operate in a given context [4], but, and this is seminal in our view, without disambiguating cognition from the body’s action in reality (e.g., Cooke and Lawless conclude that an individual’s level of intelligence outside of an interaction does not improve the interaction, but that the relevant intelligence arises, and is embodied, in the interaction itself [16]).

In Table 1, we consider interdependence historically, mindful of its usability by Artificial Intelligence (AI). Based on our historical approach, the literature, and our own research, we define interdependence as embodied cognition ([15,17]), not only entangled with reality, but also unable to be easily disentangled. We focus our article on the three effects arising from the phenomena of interdependence: it is bistable (e.g., two-sided interpretations of reality; debates; checks and balances in governance); it creates a measurement problem (e.g., questionnaires built around the notion of convergence into a single concept increase uncertainty by reducing the likelihood of perceiving the bistable, non-converged, opposing, countering aspects of reality); and non-factorability (e.g., debates, disagreements and fights are common, but who is right or who has won cannot be said without an impartial judge, jury, audience or metric to determine the outcome; we consider non-factorability to be the key characteristic that aligns teams with quantum-like (QL) entanglement).

### 1.2. Background: Justification for the Rejection of Spinoza and Hume

Until now, most social science research has focused on the individual in the laboratory, not the interaction; but in the laboratory, the individual is not governed by interdependence. While admitting to the ubiquity of interdependence in human affairs, in 1998, the leading social psychologist, Jones (p. 33, in [13]) concluded that the study of interdependence in the laboratory led to effects that were “bewildering”, sidelining the study of interdependence for a generation, unfortunately leaving the individual as the primary focus of study.

According to Spinoza [6], no causal interaction exists between bodies and ideas, that is, between the physical and the mental [18]. Whatever happens in the body is reflected or expressed in the mind. This notion by Spinoza has led to the assumption that aggregating the observed cognition of individuals subsumes individual behaviors [19], thus needing only independent data (viz., i.i.d. data; but see [20]) to improve the lives of individuals or for the betterment of teams.

In the same vein, Hume’s [7] “copy principle" holds that there is one-to-one correspondence between ideas and reality.

Let us consider individuals first. Spinoza’s and Hume’s idea has led to the development of modern measures in the laboratory for individual perceptions and beliefs that correlate strongly with other self-perceived measures. For example, self-esteem has been found to correlate significantly with other measures of mental health, leading the American Psychological Association in 1995 to consider self-esteem to be the premier goal for “the highest level of human functioning” ([21]). However, since then, and despite self-esteem’s strong correlations with other self-perceived skills, Baumeister and his team found that self-esteem in the open was not correlated with actual academic or actual work performance ([22]).

Similarly, in recent years, the concept of implicit racism has been significantly involved in driving major changes across social, academic and work relationships; however, the implicit racism concept was found to be invalid in 2009 [23]. Despite the failure of this concept, numerous training events designed to counter the “ill” effects of implicit racism have taken place, but the results have been “dispiriting” [24]. Further, a National Institutes of Health panel asked “Is Implicit Bias Training Effective”, concluding that “scant scientific evidence” existed (see NIH’s Implicit Bias Proceedings 508 at https://diversity.nih.gov/sites/coswd/files/images/NIH, accessed on 15 March 2023). Yet, Leach, the lead editor of groups in social psychology, has pushed his journal members to focus exclusively on biases [25]. But this persistence on applying concepts for individuals developed in the laboratory that subsequently fail to be validated in the open has led to the present replication crisis in the social sciences [26]. Further, the complexity in an open domain and the tasks performed there may lead to a sampling bias that negatively affects the relationship between interdependence and individual beliefs.

We are not concerned with the replication crisis, per se. We are concerned with the reason why models based on individuals seemingly cannot be generalized to the interaction or to teams.

Traditional models also include large language models like game theory and OpenAI’s ChatGPT. Strictly cognitive, for game theory, Perolat’s team [27] concluded that real-world multi-agent approaches are “currently out of reach for state-of-the-art AI methods”. In the research highlights for the same issue of *Science*, Suleymanov said of Perolat’s article: “real-world, large-scale multiagent problems …are currently unsolvable”. ChatGPT and two-person games are also assumed to easily connect to reality, but ChatGPT skeptics exist ([28,29]). Quoting from Chomsky’s [30] opinion in the *New York Times*,

OpenAI’s ChatGPT, Google’s Bard and Microsoft’s Sydney are marvels of machine learning. Roughly speaking, they take huge amounts of data, search for patterns in it and become increasingly proficient at generating statistically probable outputs—such as seemingly human-like language and thought. These programs have been hailed as the first glimmers on the horizon of artificial general intelligence …That day may come, but its dawn is not yet breaking …[and] cannot occur if machine learning programs like ChatGPT continue to dominate the field of A.I. …The crux of machine learning is description and prediction; it does not posit any causal mechanisms or physical laws.

With ChatGPT, an artificially intelligent mind uses reinforcement learning in large-language models, often making common sense errors that indicate a poor connection between the artificial mind and physical reality. Chomsky’s position mirrors that of AI researcher Judea Pearl’s conclusion when he recommended that AI researchers could only advance AI by using reasoning with causality ([31,32]). We add that reasoning about causality cannot occur with teams or systems composed of humans and machines working together without accounting for the interdependence that physically exists in the social sphere, especially among human–human teammates. Otherwise, ignoring interdependence for human–machine teams would be like treating quantum effects as “pesky” in the study of atoms.

Again, our problem is not with biases, the replication crisis, or large language models, but the results which indicate that there is little guidance to be afforded by the social science of the individual for the development of autonomous human–machine teams. This sad state changes dramatically if instead we treat the current state of social science as evidence of the measurement problem reflecting an orthogonality between the “individual” and the “team”, or between language (concepts) and action [33].

Indeed, we suggest that the state dependency [34] created from the interdependence between individuals and teammates may rescue traditional social science from its current validation crises. Simply put, if the “individual” is orthogonal to its participation in a “team”, then, by measuring the individual, evidence of the team is lost, explaining why complementarity has failed to produce predicted effects in close relationships [35].

Similarly, it may be true that language isolated from the effects of physical reality explains why ChatGPT has been criticized for its disconnect from reality; i.e., the complexity inherent in large language models impedes the possibility to “identify state variables from only high-dimensional observational data”; in [36]). In an interview, Chen, a roboticist at Duke University, expressed the folowing: “I believe that intelligence can’t be born without having the perspective of physical embodiments” (ref. [36]; also [37]; and see embodied cognition, in [15,17]). We proceed further than Spinoza–Hume and Chen by asserting that embodied thoughts derived while operating in reality cannot be disambiguated from each other. This accounts for Chomsky’s [30] conclusion that ChatGPT does not capture reality.

### 1.3. Background: Domain Complexity of a Team’s Task

Open-world and especially real-world learning has taken on new importance in recent years as AI systems continue to be applied and transitioned to real-world settings where unexpected events, such as novel events, can and do occur [38]. When designing AI systems with human–machine teams, novelties may cause additional uncertainties, a team’s performance drops, and conflict.

When designing an AI that can operate in real-world domains, we need to know about the level of complexity of the target task. The complexity level of task domain affects the interdependence of the human–machine team. Regardless of the complexity in the structure of a team, as its degrees of freedom are reduced, the perfect team operates more and more as a unit, the reason why the “performance of a team is not decomposable to, or an aggregation of, individual performances” [39]. A decomposed structure can be very complex, but while that may be, as its complex pieces begin to fit together, the degrees of freedom *reduce*, thereby reducing structural complexity. The structure’s “decomposed” complexity should match the complexity of the problem addressed; the structure, as it *unifies*, allows the unified structure the ability to produce maximum entropy (MEP). This latter part conserves “the available free” energy, i.e., the more free energy consumed by a structure to make its team “fit” together, the less free energy available for team productivity.

The complexity level of the task domain defines the skills and tools that an agent or team of agents need to perform successfully. These skills are composed of mental and physical skills. Knowing the skills needed to successfully perform a task defines the number and the combination of human and machine agents with tools needed to form a human–machine team. The complexity level of the task domain also defines the level of their interdependence in the human–machine teams, each with certain skill sets that complement a team’s mission. Further, understanding the complexity of the agents that are human and machine helps to define which agent (e.g., human or machine) with the appropriate skills and the number of them are needed to perform a task.

In other words, the complexity of a task domain defines the skills that a particular human or machine agent needs to fit with a machine or human agent’s skills; e.g., for a task that requires a team member to have good vision at night, the human-agent team needs night vision skills. However, a particular human agent who happens to be near-sighted and cannot wear corrective contact lenses (physical skill) may not be eligible to participate in a night-time task. Thus, the human-agent teams must be able to be equipped with the tools and the appropriate skills to function on a mission. Understanding the complexity of the task domain helps in transitioning from theory to simulations, from the laboratory to real-world domains; understanding the boundaries and limitations; understanding the risks for the team and agents, decreasing the uncertainties regarding fit; avoiding sampling bias; forming anticipatory thinking; defining causality in embodied thinking; and forming an understanding of tasks, mental and physical skills, and a number of agents needed to solve a problem.

Doctor et al. [10] broke down domain complexity across domains into intrinsic and extrinsic components, and each into subcomponents. Intrinsic domain complexity is where the agent performing a task does not change the complexity of the task domain. The extrinsic complexity of the domain depends on an agent’s skills; e.g., if the task domain is to lift a rock, its complexity differs for an agent that is smaller than the rock versus a larger agent that can pick up the rock or a machine that can lift a rock. Although they referred to interaction with other agents [10], they did not address the interdependence of teams with multiple agents.

Interdependence exists in the extrinsic domain complexity space. Interference reduces internal complexity when interdependence is constructive; it increases complexity when the interference is destructive; plus, destructive interference may consume all of the available free energy from a project, collapsing productivity, e.g., divorce in marriage or in business. However, the intrinsic domain complexity contributes to the skills that the agents need and the number of them, which indirectly affects the interdependence of human–machine teams. Further, redundancy increases complexity, decreases interdependence (e.g., free riding) and reduces performance [40].

### 1.4. Background: Justification for the Quantum-like (QL) Model of Interdependence

In this section, we briefly review quantum mechanics (QM); interpretations; wave functions; tradeoffs; complementarity; causality; determinism; probability; and some of the differences between classical and quantum phenomena.

The most well-known model of state dependency is quantum mechanics (QM) [34]. We introduce a brief review of the mathematics for QM, but we concentrate on establishing similarities between QM and our quantum-like (QL) model of interdependence.

The simple model of a quantum state is given by qubits. From Moskowitz [41], we consider all of the elements of $C^{2}$ of length one. That set is simply the three-sphere $S^{3}$ which is made up of elements of $v\in C^{2}$ such that ||v||=1. We call an element of $S^{3}$ under this construction a pure state. Elements $\left(\begin{array}{c} 1\\ 0 \end{array}\right)$ and $\left(\begin{array}{c} 0\\ 1 \end{array}\right)$ form an orthonormal basis of $C^{2}$. They are also elements of $S^{3}$. They are so special that they were given special names:(1)$$|0\rangle =\left[\begin{array}{c} 1\\ 0 \end{array}\right],$$
(2)$$|1\rangle =\left[\begin{array}{c} 0\\ 1 \end{array}\right].$$

Here, we begin using Dirac’s bra–ket notation for states. A mixed state is expressed by ket |ψ〉, as |ψ〉=α|0〉+β|1〉, with $\alpha ,\beta \in C^{2}$. If we also have ${\left|\alpha \right|}^{2}+{\left|\beta \right|}^{2}=1$, then our state is a pure state. If a ket is a (complex) non-zero scalar multiple of another ket, those two kets represent the same physical state.

In considering Equations (Equation 1) and (Equation 2) from the perspective of our research, we postulate that the relationship between the team and the individual is state dependent, connecting our work to quantum mechanics. In this sub-section, we elaborate on what we like about this connection. The Copenhagen interpretation of quantum mechanics led by Bohr [42] argued that quantum waves were not real, but that these waves reflected an observer’s subjective state of knowledge about reality. In Bohr’s Copenhagen interpretation, the wave function is a probability that collapses into a single value when measurement produces an observable. In the Heisenberg and Schrödinger model, canonical conjugate variables form mathematical relations (e.g., position or momentum; time or energy). But Bohr’s [43] later theory of complementarity is his generalization of what we construed as tradeoffs existing between orthogonal perspectives common in ordinary human life (e.g., [44]).

*Interpretations*. This is where belief logic, or disembodied languages, fail [2]. While Chatbot or intuition is unable to address causality [31,32], and while quantum logic works well, it too is unable to provide a consensus regarding interpretation or meaning [45].

For example, at the very start of their recent book, *Oxford Handbook of the History of Quantum Interpretations*, the authors, Bacciagaluppi and Freire Jr. (2022) [46], describe the great success of the theory but that it remains radically ambiguous in its meaning, demanding an interpretation but one that is not forthcoming. There are many competing interpretations, all of which are unsatisfactory in some way, and it seems there will be no resolution anytime soon. However, the comments by Bacciagaluppi and Freire Jr. reflect more on the difficulties of interpreting QM and the lack of consensus concerning such interpretations.

We found it curious that in the open field, QM works well in producing predicted causal effects, but not at all “in explaining causality,” in reaching a consensus on its meaning, as concluded by Weinberg [45], despite that being his stated goal. Quoting from Bacciagaluppi and Freire Jr. (p. 1, in [46]),

Quantum mechanics, created in 1925–1927, is approaching its centenary with an impressive record. It became the backbone of most research in physics, led to applications such as the transistor and laser, and prompted an upheaval in the philosophy of science …This century of conquests has also been a time of ongoing debates about the foundations and interpretation of the theory”.

Applied to our QL model, we think the reason is that Chatbot, game theory and interpretations of reality are disembodied; witness that despite success, the conclusion regarding game theory’s latest success in mastering the game of Stratego was that according to Perolat and his team (p. 996 in [27]), real-world multi-agent approaches have “astronomical state spaces characterized by imperfect information, which are currently out of reach for state-of-the-art AI methods to be applied in an end-to-end method”. That is, “real-world, large-scale multiagent problems that are characterized by imperfect information and thus are currently unsolvable”.

Similarly striking, despite significant correlations with other self-reported beliefs (mental health, depression, etc.), many championed social concepts (self-esteem, implicit racism, and superforecasting) have been found to be invalid (refs. [22,23,47]; respectively), leading to the replication project in social science ([26]). Thus, we advance the pattern: cognitive beliefs abound, despite their lack of connection to reality. If AI is to succeed with machines and humans, they need to have a common reference frame other than language alone; we chose entropy.

Yet, there is new evidence that we must consider in the future of the possibility of a breakthrough that combines large-language models with robots and lots of training [48]. (see “Aided by A.I. Language Models, Google’s Robots Are Getting Smart”, with video provided in the *New York Times* 2023).

There are many interpretations of the Copenhagen interpretation of quantum mechanics. Addressing many of these in his 2013 (p. 95) textbook [49], Weinberg concluded that "today there is no interpretation of quantum mechanics that does not have serious flaws, and that we ought to take seriously the possibility of finding some more satisfactory other theory, to which quantum mechanics is merely a good approximation". In 2017, Weinberg declared [45]: "it is a bad sign that those physicists today who are most comfortable with quantum mechanics do not agree with one another about what it all means".

Even Bohr changed his view several times. Bacciagaluppi and Freire review multiple presentations of Bohr’s philosophy of physics [46]. We also accept that some versions of the Copenhagen interpretation do in fact attribute a physical significance to quantum waves.

*Tradeoffs*.

From Bohr (p. 440, in [44]), there is an “arbitrariness” in that “no sharp separation between object and subject can be maintained, since the perceiving subject also belongs to our mental content”. Bohr added that "mutually exclusive situations [can be] characterized by a different drawing of the line of separation between subject and object" (p. 441 in [44]). Bohr also wrote that “the new situation in physics has so forcibly reminded us of the old truth that we are spectators as well as actors in the great drama of existence” (p. 439 in [44]).

Regarding tradeoffs, given a Fourier transform pair (time–energy, position–momentum), Cohen (p. 45, in [50]) found in signal theory that a

narrow waveform yields a wide spectrum, and a wide waveform yields a narrow spectrum and that both the time waveform and frequency spectrum cannot be made arbitrarily small simultaneously.

While presenting a convincing demonstration of tradeoffs, Cohen was addressing signals easily repeated and replicated, unlikely with human–machine teams.

In a team, we anticipate that humans and machines can see whether their team’s structure or performance is working to produce anticipated effects. If the team is structurally integrated (i.e., producing low entropy), it should be more productive, but if not, then it should be less productive. The mathematical aspect we hypothesized and found evidence to support occurs especially with mergers, which indicate that success is poor and may be random (e.g., see [51]).

The lack of structural integration with a merger can be significant; e.g., from the open marketplace, the poor integration of merged firms led to the shut down of Yellow, the trucker hurt by the mergers it sought and the debt that resulted. (From the *Wall Street Journal*, *Trucking Giant Yellow Shuts Down Operations. The 99-year-old company with 22,000 Teamsters employees advises customers and workers of shutdown:* “The two companies combined back-office functions but not networks, limiting cost savings. In 2005, Yellow bought another large competitor, USF …again combining back-office functions but not the broader company”).

*Complementarity*.

We define complementarity from the perspectives of *Psychology*, *Philosophy*, and by Khrennikov, a mathematician. From *Psychology* (p. 207, in [35]), complementary roles are dissimilar ones.

Bohr introduced complementarity in his first interpretation of QM in 1927, and it is different than merely having to do with the “orthogonal” perspective common in ordinary life. In fact, Bohr expressly said that complementarity is an artificial word that has no meaning in the ordinary language.

We are aware of the concerns about “complementarity”, but we are unsure of their implications for our QL model. Pais (p. 424, in [44]) wrote: “Bohr particularly admired William James: ’I thought he was most wonderful’”.

Regarding complementarity, William James (p. 204, in [52]) wrote: “…in certain persons, at least, the total possible consciousness may be split into parts which coexist but mutually ignore each other, and share the objects of knowledge between them. More remarkable still, they are complementary”.

From *Philosophy*

(Faye, in [53]), Bohr’s more mature view, i.e., his view after the EPR paper, on complementarity and the interpretation of quantum mechanics may be summarized in the following (we quote No. 14 from Faye’s list, in [53]):

Such phenomena are complementary in the sense that their manifestations depend on mutually exclusive measurements, but that the information gained through these various experiments exhausts all possible objective knowledge of the object.

Bohr considered the demands of complementarity in quantum mechanics to be logically on par with the requirements of relativity in the theory of relativity (Faye, in [53]). He believed that both theories were a result of novel aspects of the observation problem, namely the fact that observation in physics is context-dependent. It is impossible in the theory of relativity to make an unambiguous separation between time and space without reference to the observer (the context) and impossible in quantum mechanics to make a sharp distinction between the behavior of the object and its interaction with the means of observation (see Bohr, 1998, p. 105, in [54]).

From Faye [53], “Bohr pointed to two sets of descriptions which he took to be complementary. On the one hand, there are those that attribute either kinematic or dynamic properties to the atom; that is, “space-time descriptions” are complementary to “claims of causality”, where Bohr interpreted the causal claims in physics in terms of the conservation of energy and momentum. On the other hand, there are those descriptions that ascribe either wave or particle properties to a single object”.

From Khrennikov (2019), in [55], we can present the complementarity principle as composed of the following components:

(B1): There exists the fundamental quantum of action given by the Planck constant *h*.

(B2): The presence of h prevents approaching internal features of quantum systems.

(B3): Therefore, it is meaningless (from the viewpoint of physics) to build scientific theories about such features.

(B4): An output of any observable is composed of contributions from a system under measurement and the measurement device.

(B5): Therefore, the complete experimental arrangement (context) has to be taken into account.

(B6): There is no reason to expect that all experimental contexts can be combined. Therefore, there is no reason to expect that all observables can be measured jointly. Hence, there exist incompatible observables.

(B6) can be called the incompatibility principle; this is a consequence of (B4) and (B5). Typically, the complementarity principle is identified with (B6). However, such a viewpoint does not match Bohr’s understanding of the complementarity principle, as the combination (B1)–(B6). This is the good place to remark that (B6) is very natural”.

Here is how we plan to apply complementarity. From Wang and Busemeyer (see [56]), “Bohr borrowed the term from the psychologist, William James: Different measurement conditions for observing different phenomena are complementary when, a: they are mutually exclusive, and only one can be applied at any time; and, b: they are all necessary for a comprehensive account of these phenomena”.

*Causality*: The National Academy of Sciences [39] addresses classical causality and determinism for human–machine teams: “ML [machine learning] …has no causal mode …Because AI cannot use reason to understand cause and effect, it cannot predict future events, simulate the effects of potential actions, reflect on past actions or learn to generalize to new situations” (p. 8, [39]). Causal models are necessary to understand situation projections” (p. 30, [39]).

What is causality regarding AI? Pearl (see Pearl in [31] and Pearl and Mackenzie in [32]) argued that for AI to be successful, it has to be able to explain its decisions to humans with causal language. That Chomsky [30] considers ChatGPT to be a major advance, yet unable to reason about reality, is significant, seconded by others including Wolfram [57] who explains that it combines a large language database, a neural net, and reinforcement to form a “reasonable model” of the text that humans write.

From Weinberg (pp. 145, 198, in [58]), the commutation relation follows the “causality” condition for at space-like separations (i.e., no signal from x can reach y so that measurements at x do not interfere at point y (*x* and *y* or *x* and *x’*).

*Determinism*: “An assumption of teaming is that it is not deterministic (i.e., not akin to choregraphing a collection of autonomous agents)” (p. 53, [39]). [And yet] …military systems …assumes that the underlying decision support systems rely upon deterministic algorithms that perform the same way for every use” (p. 71, [39]).

*Probability*: From Kolmogorov [59], probability models operate well where questions have unambiguous answers in the data collected by “repetition” (p. 3).

In sets of possibilities of events, probability theory is used to evaluate the degree of risk in making a decision (pp. 14–15, in [60]). But risk is composed of perceptions and determinations ([61]). Risk perceptions can lead to tragic decisions, like the U.S. Department of Defense’s ([62]) decision in 2021 to fire a drone on a suspected terrorist, killing instead 10 civilians, mostly children.

*Classical versus Quantum Phenomena*: From Suskind and Friedman [63] on the difference between classical and quantum physics, “our ordinary intuition about physical systems is that if we know everything about a system, that is, everything that can in principle be known, then we know everything about its parts. …But Einstein explained to Bohr—in quantum mechanics, one can know everything about a system and nothing about its individual parts …” (p. xii). Determinism breaks down in QM, “but in a particular way” (p. 9); i.e., repeated measurements in QM on average can equal classical mechanics (CM), up to a point; e.g., “measuring one component of spin destroys the information about another component” (p. 13). Regarding a combined system that is separable (pp. 206–208), once we determine the probability of a combined system *P(a,b)*, if we sum over *b*, then we know the probability *P(a)*. But this is not possible if the system is entangled, not separable.

Psychology has struggled from its beginning over its dual tracks between mentation and movement, dwelling on sensory effects (signal detection theory), pragmatics (problem solving), behaviorism, humanism, cognitive science, interdependence, groups, therapy (Freud’s reality principle), biases, contexts, etc. (e.g., [64,65]). In Quantum Theory, context has been introduced, including context switches (e.g., [66]). Jointly, David Bohm and Karl Pribram (pp. 104–105; in [67]; reviewed in *Science*, “David Bohm’s unfinished revolution”, 2023) proposed that information stored in the “holographic” brain may be associated with consciousness, leading to a “manifest reality” that connects matter and the collective of others.

In sum, determining the state of a classical system relies on measurement and the assumption that states are separable. Separability is the defining characteristic of classical systems, non-separability of quantum systems. We generalize this result from QM to QL human–machine teams with the National Academy of Sciences report that the performance of a team is not “decomposable” to the individuals in the team (p. 11, [39]).

While these definitions focus on the QM formalism, we stress that our goal is to borrow where we can from QM to develop a QL model of teams, not to build a QM model.

Returning to the QM model, we quote an article in *Physics Today*:

To date, most experiments have concentrated on single-particle physics and (nearly) non-interacting particles. But the deepest mysteries about quantum matter occur for systems of interacting particles, where new and poorly understood phases of matter can emerge. These systems are generally difficult to computationally simulate [68].

Unlike this state of the experiments described in quantum physics, our research is designed to operate in the open for teammates interacting in teams, teams interacting with other teams, and systems of teams interacting with other systems.

*Interdependence*. By modeling QL as a series of complementary tradeoffs for teams, we have had success [40]. To further advance our project, we seek a stronger mathematical foundation that human–machine teams can observe, interpret, and act upon. We begin with interdependence. The National Academy of Sciences concluded that the effect of interdependence, or mutual dependence between two or more agents, likely causes a reduction in their degrees of freedom [14].

We are particularly interested in two types of entropy: Structural Entropy Production (SEP) and Maximum Entropy Production (MEP). SEP is based on the arrangement of a team; the choices of a team’s teammates; the capability of a team to work together seamlessly, to resolve its internal problems, and to allow adjustments; but this entropy production should be as low as possible to allow the application of the maximum amount of free energy available by the team to a team’s productivity. The choice of teammates is key—the only way to know that a good choice has been made is by the reduction in entropy with the addition of a new teammate, relegating the choice to a random selection. MEP is the maximum productivity output of a team; a team should want the maximum of its free energy available or as much as possible to be devoted to its productivity, to its targeted problem, to the maximum interdependence of its teammates to be fully engaged without free riding on the task at hand, all combining to increase the likelihood that for all teammates to be in the highest state of interdependence possible, it means relegating the members to orthogonal roles with minimal overlap (e.g., cook, waiter, cashier).

For example, the most powerful hurricane (MEP) has the tightest structure around the smallest eye (SEP) (see Figure 1 below; [69]; reproduced from [69] with the permission of the American Institute of Physics. See Figure 1 in [69] at https://doi.org/10.1063/1.2349743, accessed on 1 June 2023). With this model in mind, we found that the degrees of freedom associated with the structural entropy production (SEP) of a team forms a tradeoff with the team’s maximum entropy production (MEP; in [40]).

In this article, we strive to improve and to advance the science of what that means. As an example, a mathematical model of interdependence was used to study the effect of skill for a team’s state of interdependence, yet Moskowitz [70] concluded that teams increase their “interdependence to optimize the probability of the Team of multi-agents of reaching the correct conclusion to a problem that it confronts”. In addition, Reiche [71] defined work interdependence as “the extent to which performing a work role depends on work interactions with externalized labor”.

*Interdependence as a resource*. According to Jones (see p. 33 in [13]), although humans live in a sea of interdependence, his assertion that its effects in the laboratory were bizarre subsequently reduced its value as a research topic until it regained respectability with the Academy’s study in 2015 [14]. Cummings [72] reported his anecdotal finding that the better the scientific team, the more interdependent its members. Since then, we have learned that the interdependence between culture and technology is a driving force for evolution and a resource for innovation [73]. In the future, we plan to model interdependence as a constructive or a destructive interference possibly with the superposition of bistable agents by using a Hadamard gate. In such a model, the oppression of interdependence would be destructive interference. From another direction, an alert communicated among the interdependent members of a collective couples their brains, increasing their awareness of a possible danger, minimizing the communication that needs to be transmitted [74].

Social life is permeated with the effects of interdependence [13]. For our study, we arbitrarily described three effects associated with interdependence [40]: bistability (e.g., two-sided stories; multitasking; debates); a measurement problem (e.g., cognitive concepts commonly correlate strongly with each other, but not with their physical correlates, a part of the measurement problem; in [22]); and non-factorability (e.g., fights among couples are common, but who is at fault is often undecidable without an outside observer or judge). We found that non-factorability is the aspect of interdependence that more closely aligns with quantum mechanics.

Non-factorability is the defining characteristic of teams: the dependent parts of a team cannot be factored or disambiguated (in [39]). The first person to discover this phenomenon, which he named entanglement, was Schrödinger (p. 555, in [75]):

Another way of expressing the peculiar situation is: the best possible knowledge of a whole does not necessarily include the best possible knowledge of all its parts, even though they may be entirely separate and therefore virtually capable of being `best possibly known’, i.e., of possessing, each of them, a representative of its own. The lack of knowledge is by no means due to the interaction being insufficiently known—at least not in the way that it could possibly be known more completely—it is due to the interaction itself. Attention has recently been called to the obvious but very disconcerting fact that even though we restrict the disentangling measurements to one system, the representative obtained for the other system is by no means independent of the particular choice of observations which we select for that purpose and which by the way are entirely arbitrary. It is rather discomforting that the theory should allow a system to be steered or piloted into one or the other type of state at the experimenter’s mercy in spite of his having no access to it.

Schrödinger’s idea, the whole being not equal to the sum of its parts, was adopted by Lewin [3] as the founding idea of social psychology, and later by systems engineers as their founding idea [76]. Schrödinger’s idea directly links quantum entanglement and interdependence.

As Schrödinger famously said, quantum entanglement implies that “the best possible knowledge of a whole does not necessarily include the best possible knowledge of all its parts”. Schrödinger’s concepts as an entanglement of knowledge, and in fact probabilistic knowledge, is central. However, here is the problem with the knowledge for our QL model of teams. In 2015, the National Academy of Sciences [14] reported that “team cohesion is positively related to team effectiveness” but that the relationship is moderated by task interdependence such that the “cohesion–effectiveness relationship is stronger when team members are more interdependent”. Also from the Academy [14], a team consists of “…two or more individuals with different roles and responsibilities…” If different roles means orthogonal relationships, the lack of information shared by them would explain why a knowledge of complementary behavior has not yet been established: “Interdependence means that important behaviors will be highly correlated. However, the evidence for complementarity is scarce (p. 207, in [35])”.

It may be that what we want to describe as “knowledge” must have integrated “causal concepts” before we accept it as knowledge [20]; i.e., it must have dealt adequately with independent and identically distributed (i.i.d.) data, which, by definition, is separable information that does not include interdependent data. Another way of considering the problem is collecting the i.i.d. data observed for a social event cannot reconstruct the event observed, especially if the roles are orthogonal.

If, as teams become more cohesive, they lose internal degrees of freedom, that would explain why the Academy found in 2021 that the “performance of a team is not decomposable to, or an aggregation of, individual performances” (p. 11, in [39]). This last statement in 2021 by the Academy provides support for our QL theory that the information internal to entanglement and interdependence are similarly dependent. A primary indicator of whether “an arbitrary quantum state is entangled or separable” is a key question (e.g., p. 3, in [77]). We conclude that interdependence links QL behavior and Schrödinger’s concept of entanglement, which is our discovery (we repeat this part about QL in the conclusions).

*Teams*. In this study, our focus is on teams. In 2015, the National Academy of Sciences [14], citing Cummings [72], claimed that interdisciplinary scientists were the poorest of team performers, mediated by experience. We agree. An equation that we developed to account for the interdependent tradeoffs between structure and performance makes the prediction that a team struggling to achieve a coherent fit among its team members is a poorly performing team ([33,40]), supporting the claim by Cummings (see Equation (Equation 7), “The worst teams”). Also, the Academy claimed that teams enhance the performance of the individual, and our results point to the power of teams arising from a well-fitted team structure (i.e., decision advantage discussed later; in [40]).

One of our first results for teams dealt with the effects of redundancy on interdependence. We predicted and found that redundancy decreases interdependence [33]; in contrast, as interdependence increases, the cohesion of a team [72] increases and is reflected by an increase in its effectiveness: “moderated by task interdependence such that the cohesion–effectiveness relationship is stronger when team members are more interdependent …[reducing their] degrees of freedom” [14]. But, from Brillouin [78], “every type of constraint, every additional condition imposed on the possible freedom of choice immediately results in a decrease of information”. Thus, interdependent agents working together constructively (in phase) produce less Shannon information than independent agents. If correct, non-factorability means that information about the inner workings of a team are forever obscured to outsiders and to insiders, and that we must find another way to determine the best structure and best performance of teams.

We assume that a human’s brain projects various mental scenarios for the body’s actions in reality. At one extreme, the mind–body relationship for an individual is, however, more difficult to generalize to the effects of interdependence as expressed by Bohr’s theory of complementarity (but see [44]). At the other extreme, when an individual is a member of a group, we assume that the human mind becomes otherwise fully engaged with the back-and-forth occurring among a group’s members and processes. A group is also an amorphous, non-exact phenomenon that can add or lose members over time with varying impacts on the group that may be independent of the reason a group was formed (e.g., some of the qualitative reasons to join a group include “the motivation for completing personal goals, the drive to increase self-esteem, to reduce anxiety surrounding death, to reduce uncertainty, and to seek protection”, in [79]).

Not so for teams. The function of a team is to solve a targeted problem ([70]; e.g., improving a team’s productivity, effectiveness, efficiency, or quality; in [80]). A well-functioning team has the potential to raise the power of the individual or teammates in a team beyond that of the individuals independently performing the same functions but outside of a team (e.g., [14]). Thus, we assume that the individuals independently performing the actions of a team when not a member of a team serve as Adam Smith’s [81] “invisible hand,” forming a baseline that we have used in the past to determine the power of a team [1].

At this point, our concerns are three-fold: First, how to model the interdependence in a team? Second, how to model the individual as a member of a team? And third, how to measure an observable of interest for a team? With non-factorability in mind, we speculate that we can model the interference in teams with implicit waves in a state of superposition to represent interference; we can model the individual as contributing constructively or destructively to the interference in a team or faced by a team; and we use probabilities to predict the observable as the work products of a team evolve over time.

*The quantum model: Waves and particles*. From Dimitrova and Wei [82], in quantum mechanics, objects manifest as waves or particles, never as a pure particle or pure wave, always both. Whether an object manifests more as a wave or as a particle depends on a specific experiment or measurement. For example, the interference effect in Young’s double slit experiment demonstrated the wave nature of light, while Einstein’s photoelectric effect demonstrated its particle nature. The collapse or measurement combines with Born’s rule to identify the interference pattern as a probability distribution for individual detection.

If “pure” means “purely” or “only”, what is a pure particle or a pure wave? Or, for that matter, what is a particle or a wave, when neither of these concepts applies in QM in the way it does in classical physics? For our QL model, from Eagleman (p. 923, in [83]), we know that humans can interpret only one aspect of reality at a time: for bistable illusions, “…the visual system chooses only a single interpretation at a time, never a mixture”. We take this to mean that, for example, a human can perceive themself to be an individual or a member of a team at any one point in time, but not both, simultaneously.

A wave function is a vector in a complex Hilbert space which allows one the calculation of probabilities by using Born’s rule, which is necessary because probabilities are positive real numbers between zero and one. However, in Bohr’s interpretation, or, for that matter, any rational interpretation, the wave function is not a probability, but it only allows one the definition of such probabilities by using Born’s rule.

This may be a point we believe where our QL model of interdependence theory may be able to help social science. If humans can perceive only one aspect of reality at a time, it may be that concepts, such as implicit racism, are different from physical manifestations (e.g., discrimination). For example, to reiterate, after a decade of treating implicit racism, the results have been described as “dispiriting” ([24]), with “scant scientific evidence” in support of positive treatment results (see NIH’s Implicit Bias Proceedings 508 at https://diversity.nih.gov/sites/coswd/files/images/NIH, accessed on 15 January 2023), and yet, in large-scale studies, the belief is that actual discrimination still exists ([84]): “Nearly half of Americans (46%) say there is “a lot” of discrimination against Black people.” However, it may be that some concepts are insufficiently tethered to reality (e.g., disembodied).

Waves introduce superposition which allows us the aggregation of the contributions of a team’s members by adding constructive interference or subtracting destructive interference. From Zeilinger [85],

[T]he superposition of amplitudes …is only valid if there is no way to know, even in principle, which path the particle took. It is important to realize that this does not imply that an observer actually takes note of what happens. It is sufficient to destroy the interference pattern, if the path information is accessible in principle from the experiment or even if it is dispersed in the environment and beyond any technical possibility to be recovered, but in principle still “out there”. The absence of any such information is the essential criterion for quantum interference to appear.

No interference pattern exists before measurements. What Zeilinger actually says is a standard point concerning the double-slit experiment, which has two possible setups with different, complementary outcomes. One observes the “interference” effects, composed of discrete traces of the collisions between the quantum objects considered and the screen in the double-slit experiment in the corresponding setup, when both slits are open and there are no means to know through which slit each object has passed. The absence of what this means is what would destroy the possibility of the interference pattern, predicted by using the superposition of the formalism. Hence, indeed, as the quotation from Zeilinger says, “the absence of any such information is the essential criterion for quantum interference to appear”. It has nothing to do with the existence of superposition, which is a feature of the formalism and is always present. Alternatively, if such a knowledge is possible, one observes a discrete set of random, rather than interference-like, effects.

For the purposes of our QL model, applying Zeilinger to our thinking about an individual who becomes a member of a team, superposition models the interference between these two states. Measurement will “destroy the superposition and force us to one of the canonical basis states”, but until then, the “absence of any such information is the essential criterion” for the existence of superposition. For example, we found that redundancy produces destructive interference, reducing the effectiveness of a team [33]. From the National Academy of Sciences [39], the inability to factor the contributions of the individual members of a team is the Academy’s seminal finding: The “performance of a team is not decomposable to, or an aggregation of, individual performances” (p. 11, in [39]). Thus, factoring a team into its individual members ends the state of interdependence; however, if a team can be factored into its parts, it is not in a state of interdependence.

We are modeling the interaction with implicit waves. However, actual waves exist, too. For humans, gamma waves (>30 Hz) are a part of an inter-brain coupling (IBC) and synchronization that has been modeled with Kuramato weakly coupled oscillators [86]. Modeling IBC is crucial to a theoretical framework of the causal relations between socio-cognitive factors, behavioral dynamics, and neural mechanisms involved in multi-brain neuroscience [87]. Our plan is to build on this idea as a means to model the interdependence affecting a team.

*The quantum model: Phase*. We assume that phase is not relevant for an individual agent. But in our QL model, phase represents the effects of constructive or destructive interference influenced and instantiated by an interacting pair. When the phase is, on average, stable [88], a team’s structure is coherent; where the coherence time may be impeded or reduced by internal factors such as redundancy or vulnerability, the phase can be adjusted to coordinate with the other members of a team. We attribute the responsibility for “adjustments” to a team’s leader (e.g., a teacher; a coach; a boss).

In the past, we assumed that if the structure of the perfect team becomes a unit, by taking the limit of the operator for the logarithm of SEP, we found that predictions for human–machine teams from the theory of complementarity are observations that a machine can make (e.g., to use deception inside of a team, do not contribute to the team’s structural entropy; reduce structural entropy production (SEP) to allow the ability of the free energy available to a team to increase a team’s performance; a vulnerability becomes observable after an attack by witnessing an increase in an opponent’s SEP or a decrease in its performance entropy (MEP); and by dampening interdependence, authoritarianism decreases a team’s ability to innovate and increases its need to steal technology to be competitive (in [1,40]). Next, we begin to include operators.

*The quantum model: Operators*. An operator (if one is not familiar with operators in the Hilbert space, treat them as a complex valued matrix) connects a wave function, |Ψ〉, with an observable. Operators infer the linear superposition of states, i.e., the effects of the interference from two or more states. An operator evolves one state in time into another. Under a measurement, an operator collapses a superposition into a measurement basis. The Hermitian norm squared $\Psi^{*}\Psi$ of the wave function offers the probability that an event can be observed in a physical space. For measurements, a Hermitian operator offers a real number for any of the wave functions it can discern, that is, that are orthogonal. The eigenvalues are real and correspond to physical states. If ς is a Hermitian operator, its expectation value must be real: <ς>=<ς>*; e.g., see below for the complementary pair of SEP and MEP interconnected with free energy.

For Hermitian operators, in the simplest case, when ψ is a wave function for a particle in the (position) Hilbert space (with operators used for predicting the outcome of position measurement), Born’s rule states that the probability density function p(x,y,z) for predicting a measurement of the position at time $t_{1}$ is equal to $|\psi \left(x,y,z,t_{1}\right){|}^{2}$. Integrating over this density offers the probability or (if one repeats the experiment many times) statistics of finding the particle in a given region. One can also use projection operators, but one still needs Born’s or similar rule, such as Von Neumann’s projection postulate, to move from complex quantities of the formalism.

We generalize the ideas from Bohr, Schrödinger, and the National Academy of Sciences [14] to the interdependence between two agents, human or machine, operating together in a superposition. We assume that a human agent in interaction with another agent is in a bistable state, existing both as an individual and a member of a team, often in orthogonal roles. The superposition of individual agents or teammates, unlike independent mechanical objects in the physical world, occurs in interdependent states where two agents are dependent on each other, combining constructively or destructively to form patterns found in every social interaction (p. 33, in [13]). We propose that the measurement of superposition can be modeled with an operator that collapses the superposition. However, if the agents are operating in orthogonal roles (e.g., cook, waiter, cashier), their individual views of reality should not align.

*The quantum model: Quantum computers and communication*.

In this section, we provide an overview of quantum computers and quantum machine learning [89] for human–machine teams. This overview may offer guidance going forward on what to consider for human–machine teams with the arrival of quantum computers.

Quantum computation has attracted attention for factoring integers or finding unordered sets of data (see Shi et al. [90]).

Arute et al. [91] engineered a task to benchmark a quantum system with a low error rate, a task that was hard for a classical computer but easy for a quantum computer, finding a quantum speedup known as quantum supremacy, achieving a milestone.

Abbas and colleagues [92] approached complexity from the perspective of information geometry with measures that apply to classical and quantum models. They used Fisher information to determine the capacity of a quantum neural network. They found that quantum neural networks with a desirable Fisher information spectrum trained faster than comparable classical or other quantum models.

Shi et al. [93] modeled boson sampling, a computationally complex mathematical problem that cannot be efficiently simulated on classical computers, to demonstrate quantum algorithmic supremacy as part of a hybrid quantum classical approach to fitting a Gaussian function.

Quantum circuit models (see [94]) using convolutional neural networks may address the extreme complexity that can make classical approaches intractable.

In a review of security-sensitive applications, image classification and feature detection used in machine learning for autonomous and robotic systems may suffer from adversarial perturbations that cause misclassifications. As their approach, West and colleagues (2023) proposed that misclassifications can be prevented by using quantum adversarial machine learning systems.

In their survey, Tian et al. [95] reviewed the underlying distribution of training data to generate new data samples as part of quantum generative learning, promoting computational efficiency (already in the *Wall Street Journal* (ref. [96]: large language models could save business costs by performing tasks such as summarizing discovery documents without replacing attorneys. Likewise, there are plenty of similar jobs where costs can be saved by using generative models in medicine, computer programming, design and entertainment).

Polysemy is similar to the superposition in quantum mechanics where a quantum particle may possess several states at the same time and interact with other particles in a physical space (see Shi et al. [97]). To address the widespread uncertainty of context due to polysemy, the authors propose deep learning models integrated into quantum-inspired complex neural networks to address the text information losses caused by neglecting linguistic features of text, concluding that quantum-inspired deep neural networks may improve text classification.

Finally, in this section, overall, we agree that quantum computation holds great promise. Should quantum machines become a part of human–machine teams, we believe that the integrated team would continue to be quantum-like (QL) in nature. We believe that what we learn now with QL models may continue to be of value for the foreseeable future.

In what follows, we review what we have learned with the interdependent tradeoffs suggested by our equation that models the tradeoffs between a team’s human–machine structure and its performance; i.e., the better a team’s human–machine structure functions as a unit, the more likely its performance increases (maximum entropy production) for that structure.

Lastly, to make a “whole” team greater than the sum of its “parts”, produced by a reduction in the degrees of freedom among a team’s members [14], the following is necessary: the glue of interdependence and a profound shift in our view of social reality to include the non-factorability of embodied information [17], the search for team member fittedness, and the introduction of randomness for who and what fits into a specific team and, if that is the case, why not.

In the next section, we review the mathematics of interdependence in reverse; first for non-factorability and then bistability. Previously, we have reviewed the measurement problem in depth [40].

## 2. Interdependence: The Mathematics of Non-Factorability Leads to Tradeoffs

Two independent operators commute: [A,B]=AB−BA=0. First, if a state is simultaneously an eigenfunction of both operators *A* and *B*, the commutator must vanish. Second, if one of two independent factors are removed from an interaction, it should have no effect on the remaining factor. As examples, two Hermitian matrices commute if their eigenspaces coincide. However, when two operators are dependent, they can not commute: [A,B]=AB−BA≠0.

We assume that ς is the operator for an autonomous human–machine team’s structure that configures the team’s free energy operator, $E_{AF}$, for the team’s use of its available free energy to achieve maximum performance, symbolized by *M*. In other words, when a team’s available free energy, $E_{AF}$, is applied by the team’s structure fully to the team’s target problem, the team is producing MEP.

For interdependence, we assume that a state of superposition exists among the members of a team (or audience watching a debate). To reiterate, based on the National Academy of Sciences reports, while in a state of interdependence, the contributions of the individual members in a team cannot be decomposed [39]. Per Cummings [72], the best team science occurs with teams in the highest state of interdependence. That allows us the assumption that if a team’s human–machine structure forms into a perfect unit, its structural entropy production reduces to zero in the limit as its degrees of freedom (dof), collapse to one:(3)$$\lim_{\mathrm{dof}\to 1^{+}}\log\;\left(\mathrm{SEP}\right)=0.$$

Equation (Equation 3) tells us why Von Neumann’s proposal in 1966 for independent self-reproducing automata agents was flawed [98]. That is, the communication among Von Neumann’s automata occurred with Shannon information, making each one independent of each other, reducing their ability to communicate interdependently with each other.

The operator for structure, ς, should offer us an eigenvalue that is the team’s design for minimizing the entropy produced by its structure. Interdependently, the operator for the team’s performance productivity, *M*, should give the eigenvalue that characterizes the team’s ability to direct the maximum amount of its available free energy, $E_{AF}$, to the team’s target problem, producing MEP.

Next, the two operators being dependent on each other allows us the assumption that a tradeoff exists between the uncertainty in a team’s human–machine structure operator, ς, and the operator for the team’s productivity, *M*, allowing the achievement of a given structure of its maximum entropy production. We assume that the structure of a human–machine team’s structural operator, ς, reflects the eigenvalue it produces, and similarly that a team’s performance operator, *M*, generates an eigenvalue reflective of the MEP that a team is capable of achieving based on a given structure. Assuming that these two factors are not independent, violating one of the basic tenets of information theory (i.e., that this information is not i.i.d.; for a review, see [20]), then
(4)[ς,M]=ςM−Mς≠0. This last Equation (Equation 4) could represent an ordering effect commonly observed with questionnaires [99]. Instead, with it, we derive an uncertainty relation for a team between the interdependent factors SEP and MEP, creating a tradeoff between these two operators. Assuming that this tradeoff is between the complementary parts of a team (viz., not composed of independent factors) offers
(5)ΔςΔM≈C. Equation (Equation 5) states that the uncertainty in the entropy produced by a team’s operator for structure (SEP) times the uncertainty in the entropy produced by the team’s operator for productivity (MEP) is approximately constant. We used Equation (Equation 5) to link several predictions along with field results based on interdependence (best–worst teams; deception; vulnerability; risk perception versus risk determination; recovering rational choice, complexity and debate; emotion; innovation versus oppression; interdependence as a resource; non-factorability; the orthogonality of training versus education; oscillations; decision advantage; and a model of harmonic oscillation initially for a team of three agents then a team of three with a fourth redundant agent).

### 2.1. Best–Worst Teams

The best teams, organizations, and systems are reflected by Equation (Equation 6):(6)Δς→0,ΔM→∞.
The worst is reflected by Equation (Equation 7):(7)Δς→∞,ΔM→0. Equation (Equation 7) for the “worst” teams explains why divorce is potentially expensive and disruptive as a team’s structure is ripped apart by its members (in business, CBS versus Viacom; see [40]; or in a marriage, children of divorcing parents often act out [100]); its poor fittedness (fittedness is the quality of being fitted; we use it to mean the bidirectional fit between a new team member and the team) accounts for the finding by Cummings [72] that the poorest performing teams of scientists were interdisciplinary (mediated by experience; more later).

Based on Equations (Equation 6) and (Equation 7); on Christensen’s team’s findings [51] that the results of mergers on average are poor; and on the results of our own case studies [40], we hypothesize that the only observable available to human–machine insiders or outsiders is how members of a team fit together, characterized by the entropy production from a reduction in their degrees of freedom. That being the case, Equations (Equation 3) and (Equation 5) tell us that fittedness is contingent only on whether the structural entropy production drops when two, three or more teammates come together in an interaction and attempt to perform as a unit. Since disambiguation is not possible [39], random selection to seek fittedness becomes the only rational option applicable in reality (i.e., embodied rationality, mediated by experience).

### 2.2. Deception

Deception can be used inside of a team or system by a machine in a role with a hidden agenda, a double-dealer, a guise to steal intelligence or purvey harm to an organization as has happened with chatbot. A series of interviews published in the *New York Times* has served to warn about the use of chatbot in scams, to falsely accuse others, and to mislead. According to Equation (Equation 6), in performing its role at the highest level, a machine agent intent on harm or theft of intelligence should not play its position in a way that indicates anything other than that the machine is the best teammate in the role for which it is functioning until the machine, operating with its well-hidden agenda, has collected the information that it has sought.

Deception has long been a critical element of warfare. We quote from Sun Tzu (p. 168, in [101]): “Engage people with what they expect; it is what they are able to discern and confirms their projections. It settles them into predictable patterns of response, occupying their minds while you wait for the extraordinary moment—that which they cannot anticipate”.

Aldrich Ames is an example of successful espionage when, in 1985, “Ames began selling American intelligence information to the KGB. At least 10 CIA agents within the Soviet Union were executed as a result of Ames’s spying; ultimately, he revealed the name of every U.S. agent operating in the Soviet Union (after 1991, Russia)” [102].

### 2.3. Vulnerability

To discover a vulnerability in an opposing team, a human–machine team should probe its opponent’s structure. Based on Equation (Equation 7), vulnerability in a team’s opponent is characterized in one of two ways: its opponent’s structural entropy production increases; its maximum entropy production decreases; or both occur simultaneously. We present an example from Sun Tzu [103]: “Rouse him, and learn the principle of his activity. Force him to reveal himself, so as to find out his vulnerable spots”.

### 2.4. The Key Characteristic of Non-Factorability

The National Academy of Sciences 2021 report made an assertion that was not cited (p. 11, in [39]): The “performance of a team is not decomposable to, or an aggregation of, individual performances”. That the Academy’s claim was not cited tells us that the Academy liked its claim, but that it had no evidence in support. Equation (Equation 5) makes the prediction that interdependence affects the data for coherent teams by making it non-factorable, supporting the Academy’s claim. This claim and prediction need further exploration. But the Academy’s assertion is one of the first pieces of direct evidence by outsiders in support of our theory of interdependence, which is only evident with a loss in the degrees of freedom (for an interaction, a team, a system; in [14]). The key characteristic of interdependence is its reduction in the degrees of freedom among the parts of a team or system that it affects (Equation (Equation 3)), not only preventing the logical decomposition of a team, but also allowing us the claim of its similarity to entanglement. The loss in the degrees of freedom decreases the complexity of the team’s structure in exchange for an increase in the complexity of its output for the problems that a team addresses.

The Academy’s non-cited finding of non-factorability is explained above by Equations (Equation 3) and (Equation 5). As the degrees of freedom in a team are reduced, so is the Shannon [104] information about the conditions of the individual performances observable to the performers and to external observers, in agreement with Bohr’s observations of a player versus a sporting event’s observers (in [44]). More importantly, tensors can be used to model the separable elements of teams and systems, implying that when tensor products are factorable, this factorability between the members of a team indicates the non-existence of interdependence; otherwise, the members are interdependent.

Interdependent non-factorability can be represented by Von Neumann entropy, where the entropy of the whole, $S(\rho_{AB})$, is less than that of the sum of its parts, $S(\rho_{A})$ and $S(\rho_{B})$:(8)$$S\left(\rho_{AB}\right)\leq S\left(\rho_{A}\right)+S\left(\rho_{B}\right).$$

The equality in Equation (Equation 8) holds only when the parts of a whole are independent of each other, meaning that the entropy of the whole is less than or equal to the sum of the parts of the whole, validating Lewin’s [3] assertion that “the whole is more than the sum of its parts”, and the same claim by systems engineers [76], both claims that, unfortunately, are largely ignored of late [1].

### 2.5. Bistable Reality

Bistable refers to several situations in reality; an illusion that lends itself to two mutually exclusive interpretations (Figure 2). From Eagleman (p. 923, in [83]) reporting on bistable illusions, “…the visual system chooses only a single interpretation at a time, never a mixture”. Bistability also refers to an animal’s self-interest; e.g., when prey in a forest do not suspect the presence of predators, they overgraze and harm the forest [105]. Moreover, we know that individuals multitask poorly [106]; in contrast, the purpose of a team is to multitask [1].

An example of bistability that also invokes deception as a defensive maneuver to protect self-interest was given in a news account in *Science* [107]: The female partners of HIV-infected males participated in a drug study designed to prevent the transmission of HIV to females. At the end of the study, the females were asked whether they had been compliant with the drug regimen, with 95% stating that they had been compliant, indicating that the drug study had failed. However, the investigators had also taken blood samples from the female participants, which indicated a less than 26% compliance. “There was a profound discordance between what they told us …and what we measured”, infectious disease specialist Jeanne Marrazzo said.

### 2.6. Risk Perception versus Risk Determination

Reducing risk perception is a hard, complex problem using belief logic. An example of belief logic based on risk perception is the tragic drone attack approved by the DoD in August 2021. The DoD-approved drone attack in Afghanistan was against a purported suicide bomber. Instead, the attack killed 10 civilians, including children. In the after-action review of the tragedy [62], the DoD concluded that using a “red team” to challenge the DoD’s decision to launch the drone attack might have prevented the incident. Such a red-team challenge is generalizable to recovering rational choice, reducing complexity and debate.

### 2.7. Recovering Rational Choice, Reducing Complexity, and Debate

We are concerned with solutions that can be applied in open systems, a complex problem as research is moved from the laboratory simulations to the open world [10]: “agents in simulated environments navigate a much smaller set of possible states and perform deliberative reasoning search tasks over a much smaller set of possible state-action paths than what happens in the open world of a non-simulated, real environment”.

Mann [2] found that the belief logics developed in the laboratory fail in two important situations in the open: when facing uncertainty and when faced with conflict. For uncertainty, it can be addressed by circumscribing, or bounding [108], the problem [1]. For example, a military attempts to control the airspace around its battlefield; traffic-flow engineers use traffic circles to reduce uncertainty; mergers reduce uncertainty in contracting markets; and debate is managed in courtrooms. On the other hand, over-simplifying a complex domain can cause uncertainty and failure when transitioning from a simulated to the real world, which is more complex and often chaotic. Doctor and her colleagues (p. 9, in [10]) offered an example of the pathological behavior of an AI system, a robot, transitioning from a simulation to a real-world domain. The robot failed in the real world because it was designed for the much lower complexity in a simulated domain compared to the complexity of the real world. But by reducing the choices available to the robot (i.e., offering it fewer degrees of freedom), it successfully completed its mission.

Second, the value of debate among humans is to expose with an adversarial process the intrinsic uncertainty that characterizes reality, including the illusions inherent in risk perceptions, and the need to seek and test the strongest connections to reality; e.g., courtrooms address uncertainty by drawing a boundary about what can be brought before a court, discussed and demonstrated [1].

Using data dependency (caused by state dependency; in [34]), the uncertainty reduced inside of bounded spaces may recover rational choice [109], including with game theory; e.g., Simon’s bounded rationality [108]. Cross-examination in a courtroom is considered to be the greatest means to discovering truth [110], a bounded space with strict rules (judges) where opposing officers (lawyers) facing uncertainty compete to persuade an audience (jury) of each other’s interpretation of reality; legal appeals further reduce uncertainty and complexity with an “informed assessment of competing interests” [111].

Generalizing, we see that the decision of a blue human–machine team’s decisions under uncertainty on the battlefield challenged in a debate by an AI-assisted red team’s machine could prevent future tragedies by challenging perceptions of risk [62]; and why machine learning and game theory require controlled contexts, managed within a boundary. However, this model of using AI to challenge a decision with debate needs further exploration.

### 2.8. Emotion

Our plan is to model emotion as a heightened energy state that reduces the options available to decision makers, such as what occurred with the tragic drone strike in Afghanistan in August, 2021 [62], or in 1988, the shoot-down of an Iranian Airbus by the USS Vincennes in a “highly charged environment” (p. 3, in [112]). In 1992, as a pilot study [113] (in 2001, with the assistance of George Kang, Naval Research Laboratory, Washington, DC), we modeled emotion by having a volunteer read a script calmly and then reread it in an angry voice, finding that the latter angry voice produced twice the energy output of the calm voice. If energy is a scarce resource, the results indicated that the options for an emotional team narrow in a given situation. Guided by Equation (Equation 7), we speculate that a team’s struggles over its structure produce an emotional response that reduces its productivity. If we are correct, to be maximally effective with a given team or a given system, interdependence for the team must be bounded within which it is freely able to focus a team’s effort to achieve maximum performance.

### 2.9. Innovation and Oppression

We invert this concept into a question of how to impede innovation. In a study of education, freedom, and innovation [40], we found that innovation is impeded by the suppression of interdependence; e.g., by gangs, kings, authoritarians. By reducing interdependence, oppression motivates a country like China to steal innovations. In the interview by the *Wall Street Journal’s* Chief Editor [114] of General M. Hayden, the former Central Intelligence Agency (CIA) and National Security Administration (NSA) chief, Hayden stated that the Chinese stole millions of records from federal employees in the search for the innovativeness that has so far eluded China. He said that he told his Chinese counterparts:

You can’t get your game to the next level by just stealing our stuff. You’re going to have to innovate.

Oppression works by reducing interdependence and, consequently, innovation. For example, in 2018, Russia’s global innovation index was 43rd among all nations, but in 2022, it dropped to 47th (the data come from https://www.wipo.int, accessed on 6 January 2023). From the *Wall Street Journal* [115], in Russia today,

The press is now completely under state control and independent voices of dissent, like that of opposition leader Alexei Navalny, are quickly suppressed. Critics of the regime have been murdered both inside and outside the country.

China has also been charged with stealing technology. We quoted above by General Haden, the former Central Intelligence Agency (CIA) and National Security Administration (NSA) chief, that China had to raise its game. More recently, from the *New York Times* [116],

Although China publicly denies engaging in economic espionage, Chinese officials will indirectly acknowledge behind closed doors that the theft of intellectual property from overseas is state policy.

### 2.10. Orthogonality: Training and Experience versus Education

Returning to Equation (Equation 4), in 1992, an experiment was conducted in virtual reality for USAF pilots flying in air combat maneuvering. After analyzing the data, Lawless found no relationship between their performance and their repeated education by the USAF about how to solve the complex maneuvers needed to win in air combat (reviewed in [40]). Cummings [72] also reported that the adverse effects of being a part of an interdisciplinary team are mitigated by the experience of the participants, agreeing with our finding about USAF fighter pilots that the experience gained by training in the field, but not an education of air-combat maneuvering processes in the classroom, predicted superiority in air combat (Ref. [117]; reviewed in [40]).

In contrast, in a study of the nations of Middle Eastern North Africa (MENA; see the footnote below for the UN HDI data for MENA countries (https://hdr.undp.org/data-center/human-development-index, accessed on 15 November 2022)) first conducted in 2019 and replicated in 2022, Lawless [40] found a significant relationship between the education of a country’s citizens and its innovation index. By comparing these two studies, a conclusion was drawn that orthogonality existed between the physical training for fighter pilots in air combat maneuvering and the education for innovation; the former operates in the physical space, while the latter operates in the conceptual (perceptual) space. Orthogonality may be more common than considered heretofore.

If we assume that ς and *M* are operators and complex vectors in the Hilbert space, they are also eigenfunctions; if they are also orthonormal, then the dot product result is similar to the Kronecker delta:(9)$$<\varsigma ,M>=\delta_{ij}=\left\{\begin{array}{cc} 1, & \mathrm{if}\;i=j,\\ 0, & \mathrm{if}\;i\neq j. \end{array}\right.$$

Orthogonality in Equation (Equation 9) occurs only when i≠j; alignment or agreement occurs when i=j. The orthogonality between education (disembodied cognitive training) and training (embodied cognitive physical training [15,17]) modeled in Equation (Equation 9) accounts for the lack of an effect from educating Air Force combat fighter pilots in air combat maneuvering in the classrooms (disembodied) versus training in the field (embodied cognition), an effect that may also account for the replication crisis in social science, which we call the measurement problem (self-esteem; implicit attitudes; risk perceptions; for a review of the replication crisis, see [26]) and may, possibly, offer a corrective, but more exploration is necessary to confirm.

### 2.11. Oscillations

Previously, we considered the case of two Federal agencies in a conflict with which the first author was asked to help to rectify (by the State of South Carolina’s Department of Health and Environmental Control (DHEC)). The conflict regarded the Department of Energy’s (DOE’s) cleanup of its nuclear waste at its Savannah River Site, Aiken, SC, and is represented in Figure 3 [118].

As an example of oscillation [40], the DOE’s high-level radioactive waste (HLW) cleanup of its tanks was stopped by a lawsuit, but was allowed a restart by the U.S. Congress (i.e., the NDAA of 2005 (see at https://www.govinfo.gov/content/pkg/PLAW-108publ375/pdf/PLAW-108publ375.pdf, accessed on 15 January 2015)). As part of a separate compromise reflected with this new law, the U.S. Nuclear Regulatory Commission (NRC) was assigned sufficient oversight to overrule DOE’s technical decisions for its HLW tank closure program. However, from 2005 to 2011, DOE proposed a plan to restart its HLW tank closures, but NRC required DOE to make another proposal. This oscillation is represented by the back and forth between Points 1 and 2 in Figure 3. The back and forth approached 7 years until South Carolina’s DHEC (the South Carolina Department of Health and Environmental Control (DHEC) is the government agency responsible for public health and the environment in the U.S. state of South Carolina (https://scdhec.gov/, accessed on 10 January 2015)) complained to DOE’s Citizens Advisory Board at its Savannah River Site (SRS) in South Carolina that DOE was in danger of missing its legally mandated milestone to restart its tank closures. A committee on the SRS-CAB proposed a recommendation (after a request to intervene by DHEC, the recommendation was drafted by the first author, who was formerly on the SRS-CAB, but he was not a member at that time) approved by the full CAB in which the citizens demanded of DOE and NRC in public that the two agencies settle their differences and immediately restart tank closures, which happened.

In Equation (Equation 10), if we let $\left(\begin{array}{c} 1\\ 0 \end{array}\right)$ represent the adverse belief of NRC in Equation (Equation 10), the change in its belief is reflected by an orthogonal rotation of 90 degrees in Equation (Equation 10) (i.e., after matrix multiplication, we let θ→π/2) (see Figure 3):(10)$$\left(\begin{array}{cc} \cos\theta & -\sin\theta \\ \sin\theta & \;\cos\theta \end{array}\right)\left(\begin{array}{c} 1\\ 0 \end{array}\right)=\left(\begin{array}{c} \cos\frac{\pi }{2}\\ \sin\frac{\pi }{2} \end{array}\right)=\left(\begin{array}{c} 0\\ 1 \end{array}\right).$$

### 2.12. Decision Advantage (DA)

Previously, we modeled the oscillations in a debate (Figure 3) with a simple LRC-like electrical circuit that oscillates back and forth, with the audience providing resistance, causing the oscillations to stop when a decision was made; in this LRC-like model, beliefs are modeled as being a part of imaginary space, driving the oscillation of information for the benefit of the audience. Based on the rotations that occur as a debate’s representatives argue for and against a motion, the oscillations back and forth are represented by a “torque”, symbolized by τ, in the minds of the audience that drives their processing of the information, allowing us the modeling of decision advantage, DA [40]:(11)$$DA=\tau_{A}/\tau_{B}.$$ Equation (Equation 11) means by DA that one team, team *A*, was quicker than another team, team *B*, in driving the oscillations back and forth between competitors during a debate; that one team’s grasp of the complex issues was more forceful; or that one team’s perception of the eventual solution was held with more conviction than the other; etc.

By DA, Equation (Equation 11) has support in the literature and the field. From the office of the Director of National Intelligence in 2015 (pp. 6–9, in [120]),

strategic advantage is the ability to rapidly and accurately anticipate and adapt to complex challenges …the key to intelligence-driven victories may not be the collection of objective ‘truth’ so much as the gaining of an information edge or competitive advantage over an adversary …one prerequisite for decision advantage is global awareness: the ability to develop, digest, and manipulate vast and disparate data streams about the world as it is today. Another requirement is strategic foresight: the ability to probe existing conditions and use the responses to consider alternative hypotheses and scenarios, and determine linkages and possibilities. …Secrecy, however, is only one technique that may lead to decision advantage; so may speed, relevance, or collaboration.

The purpose of a decision advantage in combat is to “exploit vulnerabilities” (see p. 7 in [121]). Speed and quality decisions are important in business, too [122]. The same is true for advertisements that promote athletic performance (e.g., [123]).

## 3. Discussion

In this review of our quantum-like (QL) model, we did not focus sufficiently on boundaries, ethics and evolution, but we are mindful of these topics, and we believe them to be central to the establishment of a team’s or system’s autonomy, our overarching goal. We also spent no time in reviewing whether beliefs lead or follow actions, and vice versa, but we suspect that both occur during an interaction. As part of our exploration, by following stock market futures and prices during the day, it becomes apparent that they exchange leadership several times during a trading day. Instead, we suspect a superordinate in the background at play, nurtured by interdependence, similar to Ibn Khaldun’s (p. xi, [124]) “asabiyah” (group solidarity). We have been noticing for some time that, by suppressing interdependence with censorship and physical threats, gang-infested areas and the governance of a nation by authoritarians place both at a disadvantage (Russia’s surprise military difficulties in Ukraine, unexpected by insider and outsider observers; in [125] (“President Vladimir V. Putin’s war was never supposed to be like this. When the head of the C.I.A. traveled to Moscow last year to warn against invading Ukraine, he found a supremely confident Kremlin, with Mr. Putin’s national security adviser boasting that Russia’s cutting-edge armed forces were strong enough to stand up even to the Americans” [125])), which remains a part of our consideration.

## 4. Conclusions

Research tested in the open field in complex environments is critical to advance the science of interdependence for human–machine teams, systems, and autonomy (ref. [39]; e.g., [10]). We speculate that disembodied cognition used in the laboratory is at the root of the replication crisis in social science [26]. We believe that including interdependence, as difficult as it is to manage in the laboratory (p. 33, in [13]), should begin to remedy the problem.Pearl ([31,32]) requires AI researchers to include contact with reality in their models. However, embodied thoughts ([15,17]) derived while operating in reality cannot be decomposed from each other. This accounts for Chomsky’s [30] conclusion that ChatGPT cannot capture reality, and it makes satisfying Pearl’s demands more difficult in the laboratory alone.One barrier to an AHMT (autonomous human–machine team), noted by the National Academy of Sciences [39], was that most proposals for the design and operation of AHMTs are based on laboratory results that do not work in the real world. Lawless and colleagues [1] add that team science has been hindered by relying on observing how “independent” individuals act and communicate (viz., i.i.d. data; [20,104]), but independent data cannot reproduce the interdependence observed in teams [14].Digital communication is based on Shannon’s [104] information theory (e.g., entropy, channels, errors). While the information communicated between individuals works well based on Shannon, that information is factorable (i.e., i.i.d. data; see [20]); early on, it was recognized that interdependence in teams in the laboratory was not rational, leading to the recommendation to reduce it (e.g., [126,127]). Surprisingly, disembodied, factorable, and rational beliefs fail outside of the laboratory, even for simple concepts like “self-esteem” [22] or “implicit racism” [23], creating the replication crisis in social science [26] and an enormous waste in the effort to “treat” biases (e.g., [24]). By being disembodied [37], chatbot is not connected to reality [30]. In an interview in the *New York Times*, Joshua Bongard, a roboticist, states that “the body, in a very simple way, is the foundation for intelligent and cautious action” [128]. Unless they are walled off, restricted or bounded (e.g., [10,108]), these failures extend to disembodied, factorable, separable and rational beliefs, especially when confronted in the open by uncertainty or conflict [2].

Aspect, Clauser and Zeilinger (p. 1, from [129]), winners of the 2022 Nobel prize in physics, stated the following: “That a pure quantum state is entangled means that it is not separable …being separable means that the wave function ψ(x,y) can be written ψ(x)ψ(y).” From the Wikipedia article on quantum entanglement (“Meaning of entanglement”, https://en.wikipedia.org/wiki/Quantum_entanglement, accessed on 15 June 2023), “an entangled system is defined to be one whose quantum state cannot be factored as a product of states of its local constituents”. Since the contributions of individual members of a team cannot be decomposed, they are not separable nor factorable but dependent.

We reprise our earlier comment about “knowledge”. We agree that the science of QM is real, actual, reproducible and generalizable; that result is what we seek as we formulate our QL model. In 2015, the National Academy of Sciences [14] reported that “team cohesion is positively related to team effectiveness” but that the relationship is moderated by task interdependence such that the “cohesion–effectiveness relationship is stronger when team members are more interdependent”. As also mentioned by the Academy [14], a team consists of “…two or more individuals with different roles and responsibilities …” If different roles form orthogonal relationships, it also means that the lack of information that is able to be shared by them would explain why a knowledge of complementary behavior could not be established: “Interdependence means that important behaviors will be highly correlated. However, the evidence for complementarity is scarce (p. 207, in [35])”.

It may be that what we scientists want to describe as “knowledge” must have integrated “causal concepts” before we can accept it as knowledge [20]; i.e., it must have dealt adequately with independent and identically distributed (i.i.d.) data, which, by definition, is separable information that does not include interdependent data. Another way of restating the problem: collecting the i i.d. data observed for a social event cannot reconstruct the event observed, especially if the roles are orthogonal.

If, as teams become more cohesive, our QL model implies that they should lose internal degrees of freedom, that would explain why the Academy found in 2021 that the “performance of a team is not decomposable to, or an aggregation of, individual performances” (p. 11, [39]). This last statement in 2021 by the Academy provides support for our theory that the information internal to entanglement and interdependence is similarly dependent. A primary indicator of whether “an arbitrary quantum state is entangled or separable” is a key question (e.g., p. 3, in [77]). We conclude that interdependence links QL behavior and Schrödinger’s concept of entanglement, which is our discovery.

Thus, we conclude with our sketch of “a new framework” for our concept of embodied cognition as part of our QL model ([15,17]) that is remarkably similar to quantum entanglement. We know that constraints reduce information [78]. By reducing the degrees of freedom among agents in a social field [3], we cause interdependence to become a constraint. Interestingly, we know that when open-ended “knowledge” works, it reflects the absence of “surprise” [126]. We also know that embodied beliefs constructed in reality work very well, with humans making rational “dynamic adjustments” to fit reality as it changes (in the marketplace, see Lucas, p. 253, in [130]). Reflecting the non-factorable nature of embodied cognition, we know that teams cannot be decomposed [39], forming a no-copy principle for constituting teams similar to the quantum no-cloning principle (p. 77, in [131]) (We suspect that the inability to factor apart the independent, individual contributions of teams may play a part in assembly theory. In brief, assembly theory [132] attempts to establish that complexity is a signature of life; however, it overlooks that one of life’s well-fitted structures (characterized by fewer degrees of freedom, reduced structural entropy and non-factorability) transfers its available (free) energy to maximize the productivity of its structure’s function, helping it to survive (e.g., to be an effective structure able to pump across a wide range of activities, a heart must be efficient; p. 2326 in [133]). This was also overlooked by Von Neumann in his theory of self-reproducing automata [98]). In the open field, where embodied cognition reigns, a state of maximum interdependence was found to be critical to the best-performing scientific teams [72]; and we found that oppressive societies significantly reduced interdependence and the freedom to pursue education and innovation [40]; in every society, freedom best allows a society the marshalling of its available free energy against the problems it has targeted [70]. We thus speculate that interdependence, which is embodied cognition and cannot be factored (e.g., [14,39]), is the reason why debate is central to the open and free societies that evolve compared to those societies that stagnate, regress or de-evolve [1]. But for future research, we leave open this question: how do we create a new statistics based on the reduction in or the absence of the Shannon information?

## 5. Future Directions—Harmonic Oscillators

The future direction that we are exploring is to replace Equation (Equation 10) derived from an LRC-like simple model of a debate (debater 1, counter-debater 2 and an audience) [40] with three-coupled classical harmonic oscillators and then three-coupled quantum qutrit harmonic oscillators to model and monitor phase shifts in a three-way interaction between intelligent members of a human–machine team. Assuming that exchanges between elements of a team must always be in phase, phase shifts become important when teammates are unable to coordinate their interactions with one another. In a three-person restaurant, the three agents operating together as part of a team inside of a bounded space (cook, waiter, cashier) when an interaction has to be adjusted due to a mistake or complaint of service cause destructive interference. If a competent outsider can observe the problem occurring (viz., the restaurant’s owner), a simple nudge [134] may correct the problem. In our model, we construe this nudge as an example of phase control that may be sufficient to adjust the coupled harmonic oscillators of a human–machine team to recover a lagging phase (for the equations of interest, see [135]).

In Table 2, we contrast the models reviewed and, briefly, the benefits and weaknesses of each.

### Harmonic Oscillators—A Test of Models

The first test of our QL model will attempt to replicate our first reported finding with the theory of interdependence. In 2017, social network theorists predicted that redundancy made a network more efficient [136]; further, the National Academy of Sciences in 2015 reported that while teamwork made an individual more effective, they also speculated that “many hands make light work” [14]. Our first prediction contradicted both social network theorists and the Academy by predicting and finding that redundancy in a team reduces the effectiveness of interdependence in a team or organization ([137]). For example, at that time, we found that Exxon had one eighth as many employees as Sinopec, yet both companies produced the same amount of oil. We replicated this finding in a study of the largest militaries in the world [138].

We predict that by adding a fourth, redundant teammate to a QL team of three well-functioning agents, modeled by four qutrit harmonic oscillators, when the fourth oscillator does not contribute to the performance of the other three, the team will become less effective.

In summary, eventually, we want to have three QL models operating simultaneously: the quantum qutrit model of an intelligent three-member human–machine debate team; an opposing qutrit model; and a qudit model of the same but along with an audience. Interdependence suffuses all of the mathematics in our models. With these tools, we established the value of interdependence. It is central to debate and decision advantage, and to innovation and evolution, and to much more yet to be discovered.

## Figures and Tables

**Figure 1 entropy-25-01323-f001:**
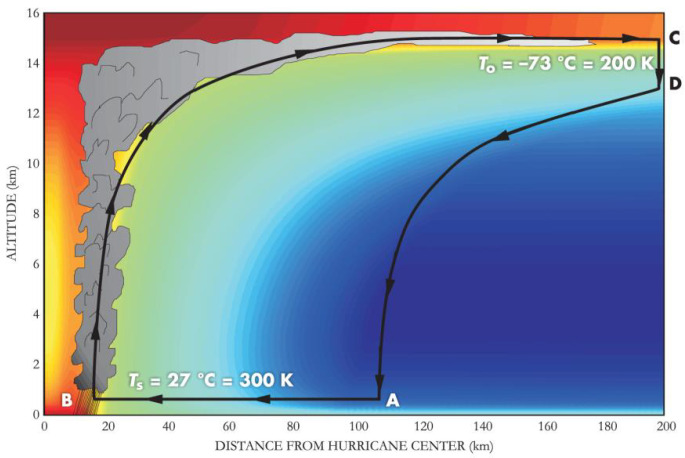
Figure 1 (from [40]). The hurricane as a simple 4-stage Carnot heat engine. The cross-section of the thermodynamic cycle to the center (upwards from B) shows the cross-section of a hurricane’s structure as a heat engine, with bright yellow–red colors in the center (maximum entropy ascending), cooler colors away from the eye-wall’s structure going outward at lower entropy. Driving the storm is the evaporation of seawater as air spirals inward (A-B), transferring energy to the air and acquiring entropy at a constant temperature. Then, an adiabatic expansion occurs, ascending inside of the eye wall until the top where entropy is dumped into the upper atmosphere (B-C). Far from the storm’s center (C-D), IR radiation exports to space the remaining entropy gained from the sea, the cooler air falling downward (D-A), compressing the fluid almost iso-thermally, followed by adiabatic compression to restart the cycle.

**Figure 2 entropy-25-01323-f002:**
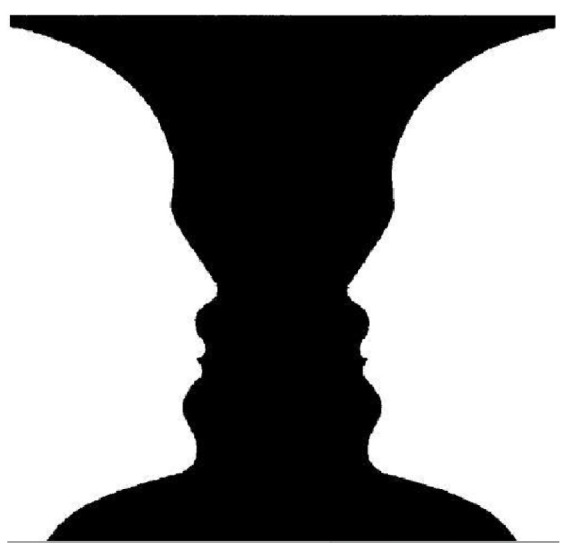
The two faces–candlestick illusion.

**Figure 3 entropy-25-01323-f003:**
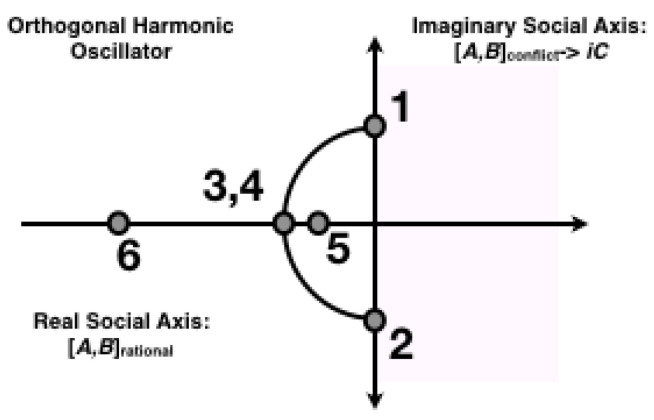
In this figure (from [40]), the *y* axis or imaginary axis represents the beliefs held not in physical reality, but imagined or constructed in the minds of debaters and re-constructed in the minds of their audience. When no audience is present and the debate is endless, the debate is located at Points 1 and 2 as a “war of words” [119] located in imaginary space. The *x* axis represents physical space or action. Encouraged by observers (e.g., an audience, jury, judge, time limit), a compromise for action is represented by Points 3 and 4. When debate is stopped by resistance from the audience with a decision often acceptable to a majority in the audience, that decision is reflected by Points 5 and 6 as a function of the resistance applied.

**Table 1 entropy-25-01323-t001:** Interdependence: A historical and computational perspective.

Authors (Year Published)	Definition	Issues
Lewin (1942) (republished [3])	Behaviour, dx/dt, occurs from the “totality of coexisting and interdependent forces in the social field that impinge on a person or group and make up the life space” (computational context; see [4]). Interdependence includes all effects that co-vary with the individual, offering an “interdependence of parts …handled conceptually only with the mathematical concept of space and the dynamic concepts of tension and force” (p. xiii) that “holds the group together …[making] the whole …more than the sum of its parts” (p. 146).	A theoretical rather than a working construct.
Von Neumann and Morgenstern, 1944 (p. 35, in [5])	“The simplest game …is a two-person game where the sum of all payments is variable. This corresponds to a social economy with two participants and allows both for their interdependence and for variability of total utility with their behavior …exactly the case of bilateral monopoly”.	Assumed that cognition and behavior are the same, like Spinoza [6] and Hume [7], and missed Nash’s solution that an equilibrium existed in a bounded space.
Nash 1950 [8]	In a two-person game, “a strategy counters another if the strategy of each player” occurs in a closed space until it reaches an “equilibrium point”.	Behavior is inferred [9]; fails facing uncertainty or conflict [2], but may be recoverable with constraints (see [1]; also, see [10]).
Kelley, Holmes, Kerr, Reis, Rusbult and van Lange [11]	“Three dimensions describe interdependence theory: mutual influence or dependence between two or more agents; conflict among their shared interests; and the relative power among them. Interdependence is defined based on specific decomposition of situations (e.g., with abstract representations such as the Prisoner’s Dilemma Game) to account for how mutual influence affects the outcome of their interactions to reach an outcome”.	After working with game matrices for decades, Kelly capitulated, complaining that situations, represented by a given (logical) matrix of outcomes, was overwhelmed by the effective matrix, which, he concluded, was based on an interaction and the unknowable. For a given game’s structure, Kelley concluded that “the transformation from given to effective matrix is subjective to observers and subject to their interpretation error with no solution in hand” [12], which Jones said caused “bewildering complexities” in the laboratory (p. 33, [13]).

**Table 2 entropy-25-01323-t002:** Table of Models.

Model	Benefit	Weakness
LRC-like model	Established Decision Advantage	Adjustments for errors are not rational.
Classic Harmonic Oscillator	Can be adjusted	Assumes that cognition and behavior form a Spinoza–Hume type monad (1:1).
Quantum Harmonic Oscillator	Can be adjusted	Assumes beliefs and behaviors form complex interactions, orthogonal under convergence (e.g., competition), parallel in mundane situations.
Qutrit Harmonic Oscillator	Can be adjusted	Allows a model of the interdependence between beliefs and actions for intelligent outsiders and intelligent insider team members (e.g., a coach).

## Data Availability

All data for this research project is open source and reported in this article.

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
