# Peer review of "A Quantum-like Model of Interdependence for Embodied Human–Machine Teams: Reviewing the Path to Autonomy Facing Complexity and Uncertainty"

_entropy, 2023, doi:10.3390/e25091323_

Round 1
Reviewer 1 Report
This review is aimed to design and test quantum-like algorithms for Artificial Intelligence (AI) in open systems to structure a human-machine team to be able to reach its maximum performance. The work is interesting, but the presentation is too abstract to understand. It is better to use more mathematical signs to describe the meaning of some definitions instead of only words. For example, when describing the quantum model, mathematical equations are necessary to describe waves and particles, phases, and operators. In addition, more detailed technical methods are suggested to be reviewed, not just definitions and basic ideas in words.
It needs more formal explanations.
Reviewer 2 Report
Unfortunately, I am compelled to advise against publication of the article. The article presents an appealing and commendable project, or more accurately a proposal for a project, rather than the completed research with establish findings, as would be preferable. As it is, the article looks more like an extended proposal for a future research project. In fairness, the authors do report some (preliminary) findings, in correspondence with their aim of modeling team-independence on quantum entanglement and related concepts, physical and mathematical, of quantum phenomena and quantum mechanics (QM). In this regard, the article does deal with quantum-like mathematical modeling in social sciences, in accord with this special issue. These findings, however, do not appear to me sufficient to merit publication. The main problem of the article, however, is the authors’ competence in quantum theory, or rather an apparent (on the basis of the article’s presentation) lack of this competence, which would undermine readers's confidence in the authors' ability to adequately use QM, as a model. Below, I list some (only some) specific problems, several of them quite severe, of the article’s presentation of QM.
p. 5 “It is interesting that Chatbot or intuition is unable to address causality [31,32], but quantum logic works well, yet quantum logic is unable to provide a consensus interpretation or meaning [39].”
First, there is a terminological issue, because “quantum logic” is a particular area of research in quantum theory, while the authors appear to mean the logic of QM in general. The meaning of the sentence is not properly explained here, especially the second part, and it is not made apparent from the subsequent discussion. In what sense QM works well in explaining causality and in what way, given that it is a probabilistic theory? What is causality? Why is QM unable to provide such a consensus? S. Weinberg does explain this in his article, even though this is a popular article, and as such not the best source for an argument in a scientific paper. This problem of over reliance of popular sources, such as NY times, is manifested throughout the article.
p. 6, ll. 238-245 “The Copenhagen interpretation of quantum mechanics led by Bohr [42] argued that quantum waves were not real, but that these waves reflected an observer’s subjective state of knowledge about reality. In Bohr’s Copenhagen interpretation, the wave function is a probability that collapses into a single value when measurement produces an observable. In the Heisenberg and Schrödinger model, canonical conjugate variables form mathematical tradeoffs (e.g., position or momentum; time or energy). But, Bohr’s [43] later theory of complementarity is his generalization of the trade offs existing between orthogonal perspectives common in ordinary human life (e.g., [44]).”
This is at best far too loose, and in fact most of this is just incorrect. First, there is no single Copenhagen interpretation, as even Bohr changed his view several times. Some versions of the Copenhagen interpretation do in fact attribute a physical significance to quantum waves, a concept that is never explained in the article either. What are quantum waves vs. classical waves? More detrimentally, what does it mean to say that the probability collapses into a single value? Probabilities to not collapse. Some speak of the collapse of the wave function (Bohr never does, however). Even worse, a wave function is not a probability: it is a vector in a complex Hilbert space which allows one to calculate probabilities, by using Born’s rule, which is necessary because probabilities are positive real numbers between 0 and 1. (More on this below) What is the Heisenberg and Schrödinger model, vis-à-vis their initially different version of QM itself, which were proved to be mathematically equivalent and then replaced by the Hilbert space formalism. Neither version was initially connected to the uncertainty relations, discovered by Heisenberg in 1927. This is not merely a matter of history. What is this mathematical trade of, why is it a trade of, and what sense it is mathematical, say, vis-à-vis the uncertainty relations, which are physical, and how they are related? Bohr introduced complementarity already in his first interpretation of QM in 1927, and it is different than merely having to do with “orthogonal” perspective common in ordinary life. In fact, Bohr expressly said that complementarity is an artificial word that has no meaning in the ordinary language. In any event, the concept is never properly explained by the authors. Does the reader need to read A. Pais’s book cited here (ref [44]), all 500 plus pages of it, to know what is meant here?
This is, as I noted, a general problem of the article: there are a lot of thrown in references instead of necessary explanations of basic concepts, moreover, without indicating where to look for these concepts in these references. Thus, the authors say:
p. 9, ll. 374-375. “From Dimitrova and Wei [58], in quantum mechanics, objects manifest as waves or particles, never as a pure particle or pure wave, always both.”
But what is the meaning of this statement? I assume that “pure” means “purely” or “only,” for otherwise what is a pure particle or a pure wave? Or for that matter, what is a particle or a wave, neither of which concepts applies in QM in the way it does in classical physics? But most problematically, in what sense quantum objects are both, including as considered in Ref. [58]? We certainly cannot conceive of objects that are simultaneously both continuous (as waves are) or discontinuously (as particles are). So, “both” must mean something else here, but it is never explained what it means
p. 10, ll. 399-395. “Measurement will “destroy the interference pattern” that exists, but until then, the “absence of any such information is the essential criterion” for the existence of superposition,” citing A. Zeilinger.
This is perhaps the most egregiously erroneous statement, which would undermine any remaining confidence the article's readers might have had left by this point of the article in the authors’ knowledge of quantum physics. No interference pattern exists before measurements. What Zeilinger actually says is a standard point concerning the double slit experiment, which has two possible set up with different, complementary outcomes. One observes the “interference” effects, composed of discrete traces of the collisions between the quantum objects considered and the screen in the double-slit experiment in the corresponding setup, when both slits are open and there are no means to know through which slit each object has passed. The absence of what means is what would destroy the possibility of the interference pattern, predicted by using the superposition of the formalism. Hence, indeed, the “absence of any such information is the essential criterion” for observing the interference pattern, as the quotation from Zeilinger says: “The absence of any such information is the essential criterion for quantum interference to appear.” It has nothing to do with the existence of superposition, which is a feature of the formalism and is always present. Alternatively, of such a knowledge us possible one observes a discrete set of random, rather than interference-like, effects.
p. 11, ll. 438-440. “For measurements, a Hermitian operator gives a real number for any of the wave functions it can discern, that is, that are orthogonal. An Hermitian operator associates a real number for each function in a set of orthogonal functions.”
A Hermitian operator does nothing of the kind. What enables one to move from Hermitian operators in a Hilbert space over C (the field of complex number) is Born’s rule, applied to the formalism, which uses operator and wave functions. Born’s rule, essentially, uses complex conjugation or, equivalently, squared moduli of amplitudes, which always gives one real quantities. These real quantities correspond to the probability densities, from which the probabilities of quantum events are derived by integrating over these densities. In the simplest case, when is a wave function for a particle in the (position) Hilbert space (with operators used for predicting the outcome of position measurement), Born’s rule states that the probability density function p (x, y, z) for predicting a measurement of the position at time t1 is equal to
. Integrating over this density gives the probability or (if one repeats the experiment many times) statistics of finding the particle in a given area. One can also use projection operators, but one still needs Born’s or similar rule, such as Von Neumann’s projection postulate, to get from complex quantities of the formalism.
Finally, the concept of entanglement, crucial for the article’s argument, including the quotation from Schrödinger (on p. 8), is never properly explained as a quantum-mechanical concept either. It is true that, as Schrödinger famously says, quantum entanglement implies that “the best possible knowledge of a whole does not necessarily include the best possible knowledge of all its parts.” However, this concept, just as does Bohr’s complementarity, has a very complex and specific meaning in QM, which in never considered by the authors. They convert this concept into a common-sense statement, which they then try to apply to the way knowledge works in team-interactions, and it may indeed apply there. But there is no convincing evidence in the article that one can in fact use the (mathematical) formalism of quantum entanglement in this case, because it is never explained that why it should work in the same way, because the actual concept of quantum entanglement (vs. a common sense that the whole is more than the sum of its parts) is never properly considered. The authors also do not properly address Schrödinger’s concepts as an entanglement of knowledge, and in fact probabilistic knowledge, which is central.
Be it as it may as concern the possibility of this quantum-like application, the authors’ treatment of quantum physics and specifically QM is unacceptable.
The quality of English itself is fine, although the article need a better structure and a sharper sense of direction.
Reviewer 3 Report
The goal of this review article is to design and test quantum-like algorithms for Artificial Intelligence (AI) in open systems in order to build a human-computer team that can achieve maximum performance. The suggestions and questions are as follows:
1. The format of the references is confusing and it is important to maintain uniformity, including case, abbreviations or full names of journals, etc.
2. Also, newspapers (such as the famous New York Times) should be cited with care in a rigorous academic paper. Newspapers can be subjective and lack rigorous academic scrutiny.
3. The abstract is too long without drawing key conclusions and highlights.
4. Review articles demand a superior standard compared to academic articles, necessitating not only a broad perspective but also a profound understanding from the author. The clarity of facts presented in the article is commendable, yet it lacks the author's unbiased viewpoint in its delivery. Upon perusing this review, my curiosity leans towards identifying the technical or theoretical inadequacies prevailing in the field, as well as the research hotspots it encompasses. I yearn to discover the ideas or perspectives that hold promise in addressing these issues. Enlightening the readers with these crucial aspects should be the primary goal of the authors.
5. The references are not new and authoritative enough, and the following are suggested additions to the references:
[1]. J. Shi, W. Wang et al., “Parameterized Hamiltonian learning with quantum circuit,” IEEE Transactions on Pattern Analysis and Machine Intelligence, pp. 1-10, 2022.
[2]. F. Arute, K. Arya et al., “Quantum supremacy using a programmable superconducting processor,” Nature, vol. 574, no. 7779, pp. 505-510, 2019.
[3]. A. Abbas, D. Sutter et al., “The power of quantum neural networks,” Nature Computational Science, vol. 1, no. 6, pp. 403-409, 2021.
[4]. J. Shi, Y. Tang et al., “Quantum circuit learning with parameterized Boson sampling,” IEEE Transactions on Knowledge and Data Engineering, pp. 1-1, 2021.
[5]. I. Cong, S. Choi et al., “Quantum convolutional neural networks,” Nature Physics, vol. 15, no. 12, pp. 1273-1278, 2019.
[6]. M. T. West, S.-L. Tsang et al., “Towards quantum enhanced adversarial robustness in machine learning,” Nature Machine Intelligence, 2023.
[7]. J. Tian, X. Sun et al., “Recent advances for quantum neural networks in generative learning,” IEEE Transactions on Pattern Analysis and Machine Intelligence, pp. 1-20, 2023.
[8] Two End-to-End Quantum-inspired Deep Neural Networks for Text Classification, IEEE Transactions on Knowledge and Data Engineering, https://ieeexplore.ieee.org/abstract/document/9627527, vol. 35, no. 4, pp. 4335-4345, 1 April 2023
Some syntactic errors require careful revision.
Round 2
Reviewer 1 Report
The authors have made significant revisions and it is much better in this version.
The quality of english languagle is not bad.
Author Response
Thank you very much for your comments!
Reviewer 2 Report
I think that the article is improved upon revisions and is much closer to being ready for publication. There remain, however, problems that should, in my view, be addressed before it is published.
The first concern the authors’ presentation of QM, even though it is better in the new version. Thus, the authors say: ‘For example, at the very start of their recent book, Oxford Handbook of the History of Quantum Interpretations, the authors, Bacciagaluppi and Freire Jr. (2022) [79], describe the great success of the theory but that it remains radically ambiguous in its meaning, demanding an interpretation but one that is not forthcoming. There are many competing interpretations, all of which are unsatisfactory in some way, and it seems there will be no resolution anytime soon.’
This formulation is very imprecise. In what sense QM as such is “radically ambiguous in its meaning,” as opposed to being difficult to interpret or to agree on an interpretation? This claim is not explained. Or, to whom? Bohr, for one didn’t think that his interpretation was ambiguous, but argues that it resolved ambiguities, specifically by means of complementarity. Why all of these interpretations are “are unsatisfactory” in some way? Is this true or are they only unsatisfactory to some, for example, those who hold alternative interpretations? These are very different claims. In fact, Bacciagaluppi and Freire Jr. or Weinberg, both of whom, the article cites, comment more on the difficulties of interpreting QM and the lack of consensus concerning such interpretations, although both might think QM itself unsatisfactory for this reason. There are of course those, beginning with Einstein, who found QM unsatisfactory as such, and there are specific reasons why he did so in this case.
Several formulations are still inaccurate and even incorrect. Thus, one example: “The Copenhagen interpretation of quantum mechanics led by Bohr [47] argued that quantum waves were not real, but that these waves reflected an observer’s subjective state of knowledge about reality. In Bohr’s Copenhagen interpretation, the wave function is a probability that collapses into a single value when measurement produces an observable.”
As I pointed out in my original report, a) in Bohr’s interpretation or for that matter any rational interpretation, the wave function is not a probability, but only allows one to defines such probabilities by using Born’s rule. There is a correct definition below in green interpolations. The authors say referring to the text in green that in their response to my report: “We plan to adopt our replies to this reviewer’s comments are in Green in manuscript.” Apart from the fact that something is wrong in this sentence, I am not sure what is the status of these comments in green now. Some of them are just formulations simply taken from my report. Either way, the above sentence needs to be corrected, as should be several other sentences.
There are still difficulties in the authors’ discussion of complementarity, although in this regard, too, the new version is improved. Thus, according to the authors: ‘Regarding complementarity, William James (p. 204, in [85]) wrote “. . . in certain persons, at least, the total possible consciousness may be split into parts which coexist but mutually ignore each other, and share the objects of knowledge between them. More remarkable still, they are complementary.” Here is the manner we plan to apply complementarity. From the literature, “Bohr borrowed the term from the psychologist, William James (Wang and Busemeyer; see in [84]): Different measurement conditions for observing different phenomena are complementary when, a: they are mutually exclusive, and only one can be applied at any time; and, b: they are all necessary for a comprehensive account of these phenomena.”’
“From literature” need not necessarily mean that one should assume that what is stated is correct. There is no real evidence that Bohr actually borrowed even the term “complementarity” from James (who was primarily a philosopher, a pragmatist, rather than a psychologist), let alone the concept which Bohr clearly did not borrow from anyone. A. Pais’s well-known comment (cited by the authors in their reply to my first report on the article) that Bohr “admired James” and “found him wonderful” is not evidence that he borrowed the term “complementarity” from James, who never used it as a noun. Not did anyone else before Bohr, which is significant because as a noun vs. an adjective it designates a new concept. Besides, James only says “they are complementary,” but he does not say in what sense. In fact, James expressly says that complementary parts of consciousness “share the objects of knowledge between them,” which is not the case in Bohr’s complementarity, which, if applied to consciousness, would in fact precludes such a sharing. The formulation defining complementarity cited from Wang and Busemeyer is closer to Bohr, although it still does not represent Bohr’s concept as used by Bohr in QM. Of course, the authors (or Wang and Busemeyer, whose work and this article I know well) are free to use complementarity in this way for their purposes, but this is not Bohr’s concept. Wang and Busemeyer summarize this formulation from an exterior article on Bohr, but this summary is not quite in accord with that article says either. In fact, most adaptations of complementarity beyond quantum physics displace Bohr’s concept as such. On the other hand, while Bohr himself often spoke about extending complementarity beyond QM, he never developed anything beyond very tentative suggestions. The article uses some of Bohr’s quotations referring to such extensions or related epistemological subjects, but while suggestive, these, unlike what Bohr says about QM and the role of complementarity there, are not accompanied by a proper argumentation. The article of course can use these formulations for its purposes, but, as general philosophical statements (not supported even by philosophical argument)s, they are very different from Bohr’s rigorous argumentation concerning complementarity and epistemology QM.
While the above problems of the article should be addressed, they are they do not significantly affect the article’s main argument. I might add, in this connection, that the views of Spinoza and Hume are greatly simplified in the article, but that, too, does not affect its main argument that much, because it only relates to a general construction that the authors (plausibly) criticize. On the other hand, the fact the concept of causality is not really explained does affect their argument, in particular as concerns the relationships/differences between: a) causality and determinism; b) deterministic and probabilistic causality; and c) causality in classical vs. quantum physics, in relation to both a) and b). (Relativity, too, changed our understanding of causality, but this is not germane here.) This remains a significant problem, noted in my original report. In fact, the authors could have benefited from Hume’s analysis of causality, central to his philosophy, but never mentioned by the article. More importantly, however, the concept of causality needs to be properly explained, and related to their argument. Part of the difficulty here is that the article overwhelmed by references, even for a review article, and needs a sharper independent articulation of its key concepts and points.
Author Response
Thank you very much for your comments. Please check the attachment.

Round 3
Reviewer 2 Report
I think the article is further improved in the second round of revisions, and in principle it can be published as is. However, while the authors made a commendable and generally effective effort in revising the content in responding to the suggestions of the reviewers, including this one, the composition of the article became somewhat unwieldy by incorporating many additional points. I would, accordingly, urge the authors to give the article a better structure and sense of direction in preparing the final version for benefit of their readers. Their various points could, I think, be integrated more harmoniously. They might, for example, consider presenting the key features of quantum mechanics in a separate section before proceeding to their quantum-like argument. But other ways of improving its structure are possible as well.